# GP2F: Cross-Domain Graph Prompting with Adaptive Fusion of Pre-trained Graph Neural Networks

**Dongxiao He** [1]   **Wenxuan Sun** [1]   **Yongqi Huang** [1]   **Jitao Zhao** [1]   **Di Jin** [1]

## Abstract

Graph Prompt Learning (GPL) has recently emerged as a promising paradigm for downstream adaptation of pre-trained graph models, mitigating the misalignment between pre-training objectives and downstream tasks. Recently, the focus of GPL has shifted from in-domain to cross-domain scenarios, which is closer to the real world applications, where the pre-training source and downstream target often differ substantially in data distribution. However, why GPLs remain effective under such domain shifts is still unexplored. Empirically, we observe that representative GPL methods are competitive with two simple baselines in cross-domain settings: full fine-tuning (FT) and linear probing (LP), motivating us to explore a deeper understanding of the prompting mechanism. We provide a theoretical analysis demonstrating that jointly leveraging these two complementary branches yields a smaller estimation error than using either branch alone, formally proving that cross-domain GPL benefits from the integration between pre-trained knowledge and task-specific adaptation. Based on this insight, we propose GP2F, a dual-branch GPL method that explicitly instantiates the two extremes: (1) a frozen branch that retains pre-trained knowledge, and (2) an adapted branch with lightweight adapters for task-specific adaptation. We then perform adaptive fusion under topology constraints via a contrastive loss and a topology-consistent loss. Extensive experiments on cross-domain few-shot node and graph classification demonstrate that our method outperforms existing methods. Code is available at https://github.com/hedongxiao-tju/GP2F.

[1]School of Computer Science and Technology, Tianjin University, Tianjin, China. Correspondence to: Yongqi Huang <yqhuang@tju.edu.cn>, Jitao Zhao <zjtao@tju.edu.cn>.

*Proceedings of the $43^{rd}$ International Conference on Machine Learning*, Seoul, South Korea. PMLR 306, 2026. Copyright 2026 by the author(s).

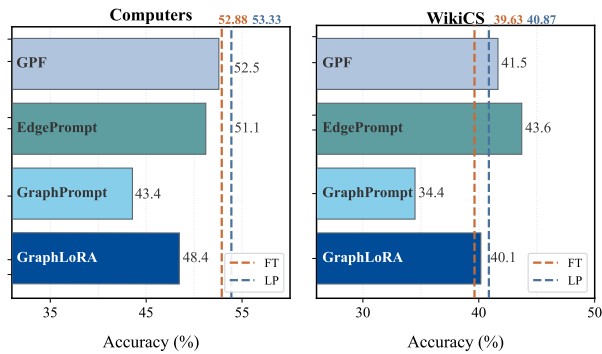

*Figure 1.* In cross-domain 1-shot scenario, where the GNN is pre-trained on *Cora* using `GRACE`, competitive performance is observed between representative GPL methods and two strong baselines, full Fine-Tuning (FT) and Linear Probing (LP).

## 1. Introduction

Graph Prompt Learning (GPL) (Zi et al., 2024; Huang et al., 2025) has emerged as an effective and promising paradigm for bridging graph pre-training and downstream tasks, reducing label reliance and alleviating the misalignment between the two stages in both optimization objectives and data distributions. Most GPLs freeze the pre-trained Graph Neural Networks (GNNs) and only train lightweight prompts during training, ensuring both efficiency and stability (Zhu et al., 2024). Moreover, GPL exhibits strong universality, covering diverse graphs such as heterogeneous graphs (Yu et al., 2024a) and text-attributed graphs (Liu et al., 2024), as well as complex downstream scenarios such as multi-task (Liu et al., 2023; Yu et al., 2024c) and cross-domain (Zhao et al., 2024; Wang et al., 2025) settings. Consequently, GPL effectively empowers various downstream applications, such as recommendation systems (Zhang et al., 2024) and biomedical analysis (Wang et al., 2024).

Recently, the research of GPL has been extended to cross-domain settings (Zhao et al., 2024; Yang et al., 2025), which is more prevalent and challenging in real-world applications, as the source and target domains differ in data distributions. For example, a model pre-trained on citation graphs may later be applied to co-purchase graphs. Such discrepancies introduce a significant gap between upstream pre-training

and downstream tasks, always leading to negative transfer across domains. To mitigate this gap, several methods have been proposed. GraphLoRA aligns the feature distributions between the source and target domains through minimizing a weighted Maximum Mean Discrepancy (MMD) (Chen et al., 2019) loss. MDGPT (Yu et al., 2024b) aligns multi-domain features via domain tokens during pre-training and adapts target domain features using mixed domain tokens. MDGFM (Wang et al., 2025) employs structure-aware tokens to reconstruct domain-invariant topology during pre-training, and aligns the target domain in both feature and structural spaces for transfer. BRIDGE (Yuan et al., 2025) trains feature aligners as experts, and activates and assembles relevant expert knowledge with a soft routing network for cross-domain transfer.

However, despite these initial successes, the effectiveness of GPL in cross-domain settings remains unclear. When a significant distribution shift exists between source and target domains, this effectiveness may stem from multiple sources: (i) general knowledge encoded in the pre-trained GNN, (ii) task-specific knowledge acquired through fine-tuning on the target domain, or (iii) implicit mechanisms that align and fuse knowledge across domains. Without a clear understanding of these contributing factors, the design of effective GPL methods often relies on empirical experience rather than principled guidance, motivating the need for a deeper theoretical investigation.

To investigate the mechanisms that make GPL effective in cross-domain scenarios, we compared representative GPLs with two simple baselines in cross-domain settings: Linear Probing (LP), which trains a linear classifier with a frozen pre-trained encoder, and full Fine-Tuning (FT), which updates all model parameters. As illustrated in Fig. 1, we observe that the pre-trained encoder achieves strong performance under both LP and FT, often matching or even surpassing existing GPL methods. This indicates that LP and FT can yield discriminative representations. To better understand the mechanism behind the experiments, in Section 3, we provide a theoretical analysis showing that, compared with using either branch alone, an appropriate combination of the two branches yields a smaller evaluation error. This provides a principled explanation for the effectiveness of GPL under domain shift, proving that fusing a frozen branch with an adapted branch can outperform either branch alone by integrating transferable pre-trained knowledge with task-specific signals.

Based on this finding, we propose GP2F, a dual-branch graph prompt learning method for cross-domain adaptation. GP2F decouples pre-trained knowledge and downstream task adaptation into two independent branches: (1) a frozen branch that preserves pre-trained knowledge using a frozen GNN, and (2) an adapted branch that introduces lightweight adapters to support task-specific adaptation under domain shift. Additionally, GP2F employs a contrastive loss to align the semantics of the two branches and further incorporates a topology-consistent loss to constrain the fusion of these two branches. Our contributions are summarized as follows:

- We find that LP and FT serve as strong baselines in cross-domain settings and provide a theoretical analysis to demonstrate that jointly optimizing the frozen branch and an adapted branch yields a smaller estimation error than using either branch alone.

- We propose GP2F, a dual-branch cross-domain graph prompt learning method that combines a frozen pre-trained GNN branch and an adapter-based adapted branch, along with a contrastive alignment loss and a topology-consistent fusion loss to jointly regularize branch-alignment and fusion.

- Extensive experiments on node and graph classification in cross-domain few-shot settings demonstrate GP2F consistently outperforms existing methods.

## 2. Preliminary

**Graphs.** We consider an attributed graph $G = (\mathcal{V}, \mathcal{E})$, where $\mathcal{V} = \{v_1, \ldots, v_N\}$ is the set of $N$ nodes and $\mathcal{E}$ is the set of edges. The node attributes are represented by a feature matrix $\mathbf{X} \in \mathbb{R}^{N \times d}$, where $\mathbf{x}_i \in \mathbb{R}^d$ denotes the feature of $v_i$. The topological structure is captured by an adjacency matrix $\mathbf{A} \in \{0, 1\}^{N \times N}$, such that $\mathbf{A}_{ij} = 1$ if $(v_i, v_j) \in \mathcal{E}$, and $\mathbf{A}_{ij} = 0$ otherwise.

**Graph Pre-training and Prompting.** We consider the graph encoder $g_\theta$, and the optimization objective during pre-training is:

$$\theta^* = \arg \min_\theta \frac{1}{|\mathcal{D}_S|} \sum_{G_i \in \mathcal{D}_S} \mathcal{L}_{\text{pre}}\big(g_\theta(G_i)\big). \tag{1}$$

where $\mathcal{L}_{\text{pre}}$ is a self-supervised objective, typically a contrastive or generative loss. $\mathcal{D}_S = \{G_1, G_2, \ldots, G_{|\mathcal{D}_S|}\}$ is the source domain dataset. $G_i \in \mathcal{D}_S$ denotes the $i$-th input graph, and $\theta$ represents the parameters of the encoder. The encoder $g_\theta$ learns generalizable knowledge by optimizing this objective. In the prompting stage, the pre-trained encoder $g_{\theta^*}$ remains frozen, and a learnable prompt module $f_\phi$ is introduced for adapting to downstream task. The optimization objective is:

$$\phi^* = \arg \min_\phi \frac{1}{|\mathcal{D}_T|} \sum_{(G_i, y) \in \mathcal{D}_T} \mathcal{L}_{\text{down}}(f_\phi(g_{\theta^*}(G_i)), y). \tag{2}$$

where $\mathcal{L}_{\text{down}}(\cdot)$ is the task-specific loss function, $\mathcal{D}_T = \{G_1, G_2, \ldots, G_{|\mathcal{D}_T|}\}$ is the target domain dataset, $G_i \in \mathcal{D}_T$ is the input graph, $y$ is the corresponding label, and $\phi$

represents the parameters of the prompt module. In this stage, the prompt module $f_\phi$ is optimized to adapt the pre-trained representations for the specific downstream task.

## 3. Theoretical analysis

We consider an anchor node $v_i$ on the target graph $\mathcal{D}_T = \{G_T\} = (\mathbf{X}_T, \mathbf{A}_T, y_T)$. Let $\mathbf{Z} = [\mathbf{z}_1, \dots, \mathbf{z}_N] \in \mathbb{R}^{d \times N}$ denote the ideal (latent) representation matrix of nodes on $G_T$, where $\mathbf{z}_i \in \mathbb{R}^d$ is the ideal representation of $v_i$ with label $y_i$. We denote by $\mathbf{h}_i^g, \mathbf{h}_i^a \in \mathbb{R}^d$ the representations produced by the frozen encoder $g_{\theta^*}$ and the adapted encoder $g_{\theta_T}$ (via lightweight adapters or prompts), respectively, and view them as two noisy observations of the same $\mathbf{z}_i$ with different error statistics.

**Assumption 3.1** (Latent linear separability). There exist latent representations $\mathbf{Z} = [\mathbf{z}_1, \dots, \mathbf{z}_N] \in \mathbb{R}^{d \times N}$ and a linear classifier $\mathbf{W}^\star = [(\mathbf{w}_1^\star)^\top; \cdots; (\mathbf{w}_C^\star)^\top] \in \mathbb{R}^{C \times d}$ satisfying Assumption A.1. There exists a margin $\gamma > 0$ such that for node $v_i$ with label $y_i$:

$$(\mathbf{w}_{y_i}^\star)^\top \mathbf{z}_i \geq (\mathbf{w}_c^\star)^\top \mathbf{z}_i + \gamma, \ \forall c \in \{1, \dots, C\}, \ c \neq y_i. \ (3)$$

This assumes that classes are separable in the latent representation, so a linear classifier can work. In practice, domain shift mainly adds noise to the representations.

**Assumption 3.2** (Second-order error model for $g_{\theta^*}$ and $g_{\theta_T}$). We assume that all expectations are taken over a randomly sampled target node $v_i$ and the randomness in training stage (e.g., sampling and optimization). Then the node representations of the frozen encoder $g_{\theta^*}$ and the adapted encoder $g_{\theta_T}$ for node $v_i$ can be written as:

$$\mathbf{h}_i^g = \mathbf{z}_i + \boldsymbol{\epsilon}_i^g, \quad \mathbf{h}_i^a = \mathbf{z}_i + \boldsymbol{\epsilon}_i^a, \quad (4)$$

where the error terms satisfy $\mathbb{E}[\boldsymbol{\epsilon}_i^g] = \mathbb{E}[\boldsymbol{\epsilon}_i^a] = \mathbf{0}$ and have bounded second moments: $\mathbb{E}[\|\boldsymbol{\epsilon}_i^g\|^2] < M, \mathbb{E}[\|\boldsymbol{\epsilon}_i^a\|^2] < M$, where $M$ denotes a finite constant. Define:

$$\sigma_g^2 := \mathbb{E}\|\boldsymbol{\epsilon}_i^g\|^2, \quad \sigma_a^2 := \mathbb{E}\|\boldsymbol{\epsilon}_i^a\|^2, \quad \rho := \mathbb{E}\langle \boldsymbol{\epsilon}_i^g, \boldsymbol{\epsilon}_i^a \rangle. \ (5)$$

We assume: $\sigma_g^2 > 0$, $\sigma_a^2 > 0$ and $\sigma_g^2 + \sigma_a^2 - 2\rho > 0$, $\rho < \min\{\sigma_g^2, \sigma_a^2\}$.

Assumption 3.2 requires that both encoders have non-zero error variance and that their errors are not perfectly aligned, i.e., the two branches exhibit partially complementary error patterns rather than behaving identically. We model $\mathbf{h}_i^g$ and $\mathbf{h}_i^a$ as two unbiased noisy observations of the same latent representation $\mathbf{z}_i$, i.e., $\mathbb{E}[\boldsymbol{\epsilon}_i^g] = \mathbb{E}[\boldsymbol{\epsilon}_i^a] = \mathbf{0}$. This zero-mean condition is a standard second-order moment assumption that allows us to focus on how the two branches differ in error variances and cross-covariance. In practice, the two branches share the same backbone, data split, and protocol,

so their deviations from $\mathbf{z}_i$ are unlikely to have a systematic directional bias. Consider their discrepancy as follows:

$$\Delta_i := \mathbf{h}_i^g - \mathbf{h}_i^a = \boldsymbol{\epsilon}_i^g - \boldsymbol{\epsilon}_i^a. \quad (6)$$

Since $\mathbb{E}[\boldsymbol{\epsilon}_i^g] = \mathbb{E}[\boldsymbol{\epsilon}_i^a] = \mathbf{0}$, we have $\mathbb{E}[\Delta_i] = \mathbf{0}$. This makes $\Delta_i$ a natural *control variate*: adding a scaled version of a zero-mean term does not change the estimator's expectation but can reduce its mean-squared error when it is correlated with the estimation error. Motivated by this, we correct $\mathbf{h}_i^a$ using $\Delta_i$ and define:

$$\tilde{\mathbf{z}}_i(\lambda) := \mathbf{h}_i^a + \lambda(\mathbf{h}_i^g - \mathbf{h}_i^a), \quad \lambda \in \mathbb{R}. \quad (7)$$

The estimator remains unbiased for any $\lambda$. Moreover, (7) is equivalent to an affine fusion:

$$\tilde{\mathbf{z}}_i(\lambda) = \lambda \mathbf{h}_i^g + (1 - \lambda)\mathbf{h}_i^a, \quad (8)$$

which in fact characterizes the general form of unbiased affine combinations of two unbiased estimates. By Assumption 3.2, we have $\mathbf{h}_i^g = \mathbf{z}_i + \boldsymbol{\epsilon}_i^g$ and $\mathbf{h}_i^a = \mathbf{z}_i + \boldsymbol{\epsilon}_i^a$. For $\tilde{\mathbf{z}}_i(\lambda)$ defined in Eq.(7), its estimation error is:

$$\tilde{\mathbf{z}}_i(\lambda) - \mathbf{z}_i = \lambda \boldsymbol{\epsilon}_i^g + (1 - \lambda)\boldsymbol{\epsilon}_i^a. \quad (9)$$

We measure the estimation error distance with latent representation $\mathbf{z}_i$ by using the mean-squared error (MSE):

$$\text{MSE}(\lambda) := \mathbb{E}[\|\tilde{\mathbf{z}}_i(\lambda) - \mathbf{z}_i\|^2]. \quad (10)$$

**Lemma 3.3** (Quadratic form of the MSE). *Under Assumptions 3.2, define $\sigma_g^2 := \mathbb{E}[\|\boldsymbol{\epsilon}_i^g\|^2]$, $\sigma_a^2 := \mathbb{E}[\|\boldsymbol{\epsilon}_i^a\|^2]$, and $\rho := \mathbb{E}[\langle \boldsymbol{\epsilon}_i^g, \boldsymbol{\epsilon}_i^a \rangle]$. Then $\text{MSE}(\lambda)$ is a strictly convex quadratic function of $\lambda$ and has a unique minimizer:*

$$\text{MSE}(\lambda) = A\lambda^2 + B\lambda + C, \quad \lambda^\star = \frac{\sigma_a^2 - \rho}{\sigma_g^2 + \sigma_a^2 - 2\rho}, \ (11)$$

*where $A = \sigma_g^2 + \sigma_a^2 - 2\rho > 0$, $B = 2(\rho - \sigma_a^2)$ and $C = \sigma_a^2$. The $\lambda^\star$ satisfies $0 < \lambda^\star < 1$ under Assumption 3.2.*

Detailed proof about Lemma 3.3 and the derivation of $\text{MSE}(\lambda^\star)$ are in Appendix A.1.

**Theorem 3.4** (Strict MSE improvement over either branch). *Under Assumption 3.2, the fused estimator at $\lambda^\star$ strictly improves over either single branch:*

$$\mathbb{E}[\|\tilde{\mathbf{z}}_i(\lambda^\star) - \mathbf{z}_i\|^2] < \min\left\{ \mathbb{E}[\|\mathbf{h}_i^g - \mathbf{z}_i\|^2], \ \mathbb{E}[\|\mathbf{h}_i^a - \mathbf{z}_i\|^2] \right\}. \quad (12)$$

Detailed proof about theorem 3.4 are in Appendix A.2. This theorem implies that if we can obtain two approximately unbiased yet non-identical estimators of the same latent signal, and learn a fusion weight close to $\lambda^\star$, then the fused representation $\tilde{\mathbf{Z}}$ achieves smaller estimation error than either branch alone. Moreover, under the margin condition

in Assumption 3.1, the misclassification probability can be upper-bounded by a constant multiple of $\mathbb{E}\left[\|\tilde{\mathbf{z}}_i(\lambda) - \mathbf{z}_i\|_2^2\right]$ (see Corollary A.2). Therefore, reducing the MSE tightens the classification error bound, providing a principled explanation for why fusing a frozen branch with an adapted branch can improve performance in cross-domain settings and motivating our method design.

# 4. Method

Guided by the theories in Section 3, we propose GP2F, a dual-branch cross-domain graph prompt learning method. For dimension alignment, we use a MLP $Proj(\cdot)$ as projector. GP2F consists of a frozen pre-trained encoder and an adapted branch equipped with lightweight adapters $\mathcal{A}$. We introduce a contrastive loss $\mathcal{L}_{\text{ctr}}$ to align the representations produced by the two branches, and a topology-consistent fusion loss $\mathcal{L}_{\text{fus}}$ to constrain the fusion.

## 4.1. Dual-Branch Encoding

**Connection to theory.** The frozen branch output corresponds to $\mathbf{h}_i^g$ and the adaption branch corresponds to $\mathbf{h}_i^a$ in Eq. (7), $\alpha$ corresponds to $\lambda$ in Eq. (8) and the goal is to obtain two approximately unbiased estimators with complementary errors.

Guided by the theoretical insight above, we design a parallel dual-branch architecture, Formally, given a target graph $G$ and the frozen pre-trained encoder $g_{\theta^*}$, the frozen branch is denoted as:

$$\mathbf{H}_{\text{pre}} = g_{\theta^*}(G). \tag{13}$$

In parallel, the adapted branch introduces a lightweight learnable adaptation module $f_\phi(\cdot)$ for task-specific adaptation, which is obtained as:

$$\mathbf{H}_{\text{adp}} = f_\phi(g_{\theta^*}(G)). \tag{14}$$

Consistent with the theoretical analysis in Eq. (8), we perform an affine fusion of $\mathbf{H}_{\text{pre}}$ and $\mathbf{H}_{\text{adp}}$ to get $\mathbf{H}_{\text{mix}}$ used in downstream objectives such as cross entropy as follows:

$$\mathbf{H}_{\text{mix}} = \alpha \cdot \mathbf{H}_{\text{pre}} + (1 - \alpha) \cdot \mathbf{H}_{\text{adp}}, \tag{15}$$

where $\alpha \in [0, 1]$ is a learnable scaling factor that adaptively weights each branch, favoring the frozen branch initially when the randomly-initialized adapted branch yields suboptimal representations, and gradually shifting to the adapted branch as training.

## 4.2. Residual Adapter with Structural Contrastive Loss

In challenging cross-domain few-shot scenarios, prior research has highlighted that aggressive parameter updates can lead to representation drift (Zhu et al., 2024). This phenomenon occurs when the model excessively distorts

the universal feature space acquired during pre-training to fit downstream tasks, thereby sacrificing the generalization priors accumulated from the source domain.

To address this issue, we employ a progressive update strategy which performs parameter-efficient fine-tuning in place of full fine-tuning. We introduce a set of learnable adapters $\mathcal{A} = \{\mathcal{A}^{(1)}, \mathcal{A}^{(2)}, \ldots, \mathcal{A}^{(L)}\}$, where $\mathcal{A}^{(l)}$ is integrated into the $l$-th layer of the encoder. Each adapter $\mathcal{A}^{(l)}$ consists of two linear layers, specifically a down-projection $\text{DOWN}^{(l)} \in \mathbb{R}^{d \times r}$ and an up-projection $\text{UP}^{(l)} \in \mathbb{R}^{r \times d}$ ($r \ll d$), with a non-linear activation in between. At layer $l$, $\mathbf{H}^{(l)} = g_\theta^{(l)}(\mathbf{H}_{\text{adp}}^{(l-1)})$ and the adapted representation is obtained as:

$$\mathbf{H}_{\text{adp}}^{(l)} = \mathbf{H}^{(l)} + \beta^{(l)} \cdot \text{UP}^{(l)}\Big(\sigma\big(\text{DOWN}^{(l)}(\mathbf{H}^{(l)})\big)\Big), \tag{16}$$

where $\mathbf{H}^{(0)} = Proj(\mathbf{X})$ and $\sigma(\cdot)$ denotes the ReLU activation function. $\beta^{(l)}$ is a learnable scaling factor initialized with a small value, which controls the adaptation intensity at each layer while ensuring the stability of the pre-trained knowledge during the initial phase of training. Furthermore, we introduce a contrastive loss $\mathcal{L}_{\text{ctr}}$, treating the representations generated by the frozen branch and the adapted branch as two distinct contrastive views, denoted as $\mathbf{H}_{\text{pre}} = \{\mathbf{h}_1^g, \ldots, \mathbf{h}_N^g\}$ and $\mathbf{H}_{\text{adp}} = \{\mathbf{h}_1^a, \ldots, \mathbf{h}_N^a\}$, respectively. For node $v_i$, the positive neighbor set is:

$$\mathcal{P}_i = \{\mathbf{h}_j^g \cup \mathbf{h}_j^a \mid j \in \mathcal{N}(v_i)\}, \tag{17}$$

where $\mathcal{N}(v_i)$ denotes the neighborhood of node $v_i$. Let $\mathcal{Q}_i$ be the set of negative samples. We define the similarity function as $\psi(a, b) = \exp(a^\top b / \tau_{\text{ctr}})$, where $\tau_{\text{ctr}}$ is a temperature parameter. The loss of node $v_i$ is given by:

$$
\begin{aligned}
&- \ell(\mathbf{h}_i^g, \mathbf{h}_i^a) \\
&= \log \frac{\psi(\mathbf{h}_i^g, \mathbf{h}_i^a) + \sum_{\mathbf{h}_j \in \mathcal{P}_i} \psi(\mathbf{h}_i^g, \mathbf{h}_j)}{\psi(\mathbf{h}_i^g, \mathbf{h}_i^a) + \sum_{\mathbf{h}_j \in \mathcal{P}_i} \psi(\mathbf{h}_i^g, \mathbf{h}_j) + \sum_{\mathbf{h}_k \in \mathcal{Q}_i} \psi(\mathbf{h}_i^g, \mathbf{h}_k)}.
\end{aligned}
\tag{18}
$$

$\mathcal{Q}_i$ is the negative set, including node pairs except $\mathcal{P}_i$. Since the two contrastive views are symmetric, both the pair $(\mathbf{h}_i^g, \mathbf{h}_i^a)$ and $(\mathbf{h}_i^a, \mathbf{h}_i^g)$ are calculated, and the total loss $\mathcal{L}_{\text{ctr}}$ is defined as:

$$\mathcal{L}_{\text{ctr}} = \frac{1}{2N} \sum_{i=1}^{N} \left(\ell(\mathbf{h}_i^g, \mathbf{h}_i^a) + \ell(\mathbf{h}_i^a, \mathbf{h}_i^g)\right). \tag{19}$$

This loss aligns the target-domain structure with the source domain, and guide the adapted branch to generate discriminative representations.

## 4.3. Topology-Consistent Fusion Loss

While the dual-branch architecture decouples universal and task-specific knowledge, $\mathbf{H}_{\text{pre}}$ and $\mathbf{H}_{\text{adp}}$ often capture distinct semantics in representation space. To mitigate the

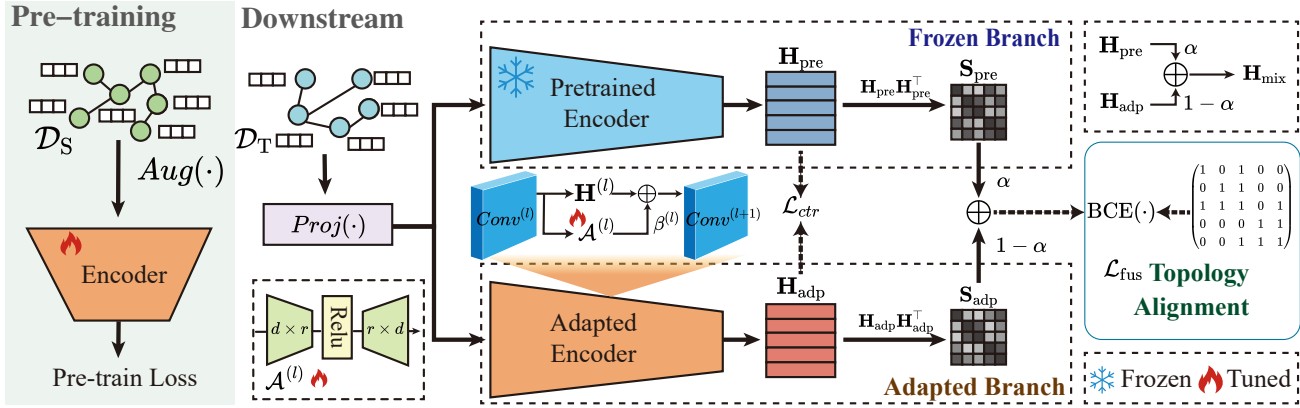

*Figure 2.* Overall framework of the proposed GP2F. Given a graph in target domain $\mathcal{D}_\mathrm{T}$ and a encoder pre-trained on source domain $\mathcal{D}_\mathrm{S}$, $Proj(\cdot)$ is a MLP used for dimension alignment. The frozen branch consists of a pre-trained GNN to preserve universal knowledge, while the adapted branch uses learnable adapters $\mathcal{A}$ for downstream adaptation. Additionally, a contrastive loss $\mathcal{L}_\mathrm{ctr}$ is used to align the two branches, and a BCE loss constrains the fusion. $\mathbf{H}_\mathrm{mix}$ is used for downstream prediction.

discrepancy between these two views, we introduce a fusion constraint based on downstream topology. We first compute the self-similarity matrices of both representations:

$$\mathbf{S}_\mathrm{pre} = \mathrm{Norm}(\mathbf{H}_\mathrm{pre}\mathbf{H}_\mathrm{pre}^\top), \quad \mathbf{S}_\mathrm{adp} = \mathrm{Norm}(\mathbf{H}_\mathrm{adp}\mathbf{H}_\mathrm{adp}^\top), \tag{20}$$

where $\mathrm{Norm}(\cdot)$ denotes the normalization operation. The mixed similarity matrix $\mathbf{S}_\mathrm{mix}$ is then generated using the learnable scaling factor $\alpha$ defined in Eq. (15):

$$\mathbf{S}_\mathrm{mix} = \alpha \cdot \mathbf{S}_\mathrm{pre} + (1-\alpha) \cdot \mathbf{S}_\mathrm{adp}. \tag{21}$$

To maintain structural consistency, nodes that are topologically adjacent should exhibit high semantic similarity in the representation space, whereas disconnected nodes should remain distant. Accordingly, we align the mixed similarity matrix $\mathbf{S}_\mathrm{mix}$ with the target topology $\mathbf{A}$. We first introduce a similarity threshold $t$ to define a consistency mask $\mathcal{M}$:

$$\mathcal{M} = \{(i,j) \mid (\mathbf{S}_{ij} > t \land \mathbf{A}_{ij}=1) \lor (\mathbf{S}_{ij} \le t \land \mathbf{A}_{ij}=0)\}, \tag{22}$$

where $\mathbf{S}_{ij}$ denotes the representation similarity between nodes $v_i$ and $v_j$ in $\mathbf{S}_\mathrm{mix}$. The topology-consistent fusion loss $\mathcal{L}_\mathrm{fus}$ is formulated via Binary Cross-Entropy (BCE) as:

$$\mathcal{L}_\mathrm{fus} = \frac{1}{|\mathcal{M}|} \sum_{(i,j)\in\mathcal{M}} \mathrm{BCE}\big(\sigma(\mathbf{S}_{ij}/\tau_\mathrm{fus}), \mathbf{A}_{ij}\big), \tag{23}$$

where $\sigma(\cdot)$ is the sigmoid function and $\tau_\mathrm{fus}$ is a temperature parameter. In practice, $\mathcal{L}_\mathrm{fus}$ is approximated by computing this loss on sampled subgraphs within each mini-batch, leading to $\mathcal{O}(B^2d)$ complexity with a small batch size $B$.

### 4.4. Optimization Objective

The overall optimization objective is defined as:

$$\mathcal{L} = \mathcal{L}_\mathrm{cls} + \lambda_1\mathcal{L}_\mathrm{ctr} + \lambda_2\mathcal{L}_\mathrm{fus}, \tag{24}$$

where $\mathcal{L}_\mathrm{cls}$ denotes the cross-entropy classification loss. The hyperparameters $\lambda_1$ and $\lambda_2$ weight the contributions of these two losses, respectively.

## 5. Experiments

In this section, we conduct extensive experiments to evaluate the effectiveness of GP2F in cross-domain few-shot learning, aiming to answer the following research questions:

**RQ1**: How effective is GP2F in cross-domain few-shot learning scenarios compared with existing baselines?
**RQ2**: Does GP2F demonstrate robustness across different pre-training strategies?
**RQ3**: How does GP2F performs compare with graph foundation models?
**RQ4**: Is GP2F effective on large-scale datasets?
**RQ5**: How do key components and hyperparameters affect the performance of GP2F?

### 5.1. Experimental setup

We evaluate the performance on nine node classification benchmark datasets including *Cora*, *CiteSeer*, *PubMed* (Yang et al., 2016), *Computers*, *Photo*, *CS* (Shchur et al., 2018), *WikiCS* (Mernyei & Cangea, 2020), *ogbn-arxiv*, and *ogbn-products* (Hu et al., 2020) and six graph classification datasets including *PROTEINS* (Dobson & Doig, 2003), *MUTAG* (Debnath et al., 1991), *DD* (Shervashidze et al., 2011), *COX2*, *BZR* (Morris et al., 2020) and *ENZYMES* (Borgwardt et al., 2005). We compare our model against four types of methods: (1) Fine-tuning (FT). (2) Self-supervised pre-training methods: DGI (Veličković et al., 2019), GRACE (Zhu et al., 2020), and GraphMAE (Hou et al., 2022). (3) Graph prompt learning baselines: GPPT (Sun et al., 2022),

*Table 1.* Accuracy of 1-shot node classification across datasets pre-trained on *Cora*. Best in **bold** and second best underlined.

| Method | Cora | CiteSeer | PubMed | Computers | Photo | CS | WikiCS |
|---|---|---|---|---|---|---|---|
| GRACE(FT) | $33.65_{\pm10.10}$ | $28.13_{\pm7.55}$ | $51.71_{\pm8.95}$ | $52.88_{\pm12.25}$ | $61.52_{\pm11.32}$ | $55.81_{\pm10.86}$ | $39.63_{\pm9.37}$ |
| GRACE(LP) | $35.53_{\pm9.68}$ | $28.47_{\pm8.44}$ | $51.97_{\pm8.91}$ | $\underline{53.33}_{\pm12.36}$ | $62.77_{\pm12.07}$ | $58.12_{\pm10.58}$ | $40.87_{\pm8.36}$ |
| DGI(LP) | $34.52_{\pm9.72}$ | $28.41_{\pm8.12}$ | $47.92_{\pm9.93}$ | $51.05_{\pm11.28}$ | $61.71_{\pm11.66}$ | $58.41_{\pm8.80}$ | $42.37_{\pm9.31}$ |
| GraphMAE(LP) | $36.32_{\pm8.91}$ | $30.70_{\pm7.98}$ | $\mathbf{53.54}_{\pm9.78}$ | $48.36_{\pm12.36}$ | $60.89_{\pm10.70}$ | $64.06_{\pm8.02}$ | $\underline{44.57}_{\pm8.13}$ |
| GPPT | $27.54_{\pm10.50}$ | $22.57_{\pm6.51}$ | $42.57_{\pm10.33}$ | $33.79_{\pm18.30}$ | $36.54_{\pm16.82}$ | $34.47_{\pm13.04}$ | $29.32_{\pm9.30}$ |
| GPF | $34.09_{\pm9.76}$ | $28.19_{\pm8.16}$ | $51.97_{\pm8.80}$ | $52.45_{\pm12.52}$ | $62.13_{\pm11.87}$ | $59.15_{\pm10.30}$ | $41.55_{\pm8.37}$ |
| GraphPrompt | $47.45_{\pm10.18}$ | $36.12_{\pm8.89}$ | $51.88_{\pm9.22}$ | $43.44_{\pm12.26}$ | $59.00_{\pm9.60}$ | $54.10_{\pm9.21}$ | $34.39_{\pm7.69}$ |
| EdgePrompt | $32.69_{\pm7.64}$ | $28.35_{\pm7.53}$ | $51.08_{\pm8.70}$ | $51.09_{\pm11.15}$ | $60.72_{\pm11.59}$ | $53.89_{\pm8.29}$ | $43.60_{\pm9.16}$ |
| DAGPrompT | $43.93_{\pm9.06}$ | $39.81_{\pm10.22}$ | $49.88_{\pm9.74}$ | $35.20_{\pm10.99}$ | $49.70_{\pm7.77}$ | $\underline{64.84}_{\pm9.31}$ | $26.50_{\pm7.72}$ |
| GraphLoRA-S | $\underline{56.25}_{\pm9.13}$ | $\underline{48.10}_{\pm10.22}$ | $52.68_{\pm12.45}$ | $48.41_{\pm10.03}$ | $\underline{66.34}_{\pm10.96}$ | $62.72_{\pm8.22}$ | $40.07_{\pm8.97}$ |
| GP2F(Ours) | $\mathbf{57.80}_{\pm8.67}$ | $\mathbf{51.55}_{\pm11.73}$ | $\underline{53.05}_{\pm10.64}$ | $\mathbf{54.89}_{\pm10.39}$ | $\mathbf{67.60}_{\pm10.93}$ | $\mathbf{68.33}_{\pm7.08}$ | $\mathbf{45.37}_{\pm8.72}$ |

*Table 2.* Accuracy of 50-shot graph classification across datasets pre-trained on *PROTEINS*. AUC denotes the AUROC and ACC denotes Accuracy. Best in **bold** and second best underlined.

| Method | COX2(AUC) | BZR(AUC) | DD(AUC) | PROTEINS(AUC) | MUTAG(AUC) | ENZYMES(ACC) |
|---|---|---|---|---|---|---|
| GRACE(FT) | $\underline{63.85}_{\pm5.95}$ | $67.04_{\pm5.77}$ | $65.46_{\pm4.19}$ | $63.30_{\pm5.50}$ | $72.44_{\pm5.31}$ | $23.50_{\pm2.47}$ |
| GRACE(LP) | $62.39_{\pm5.93}$ | $65.74_{\pm6.08}$ | $66.28_{\pm3.09}$ | $62.85_{\pm4.93}$ | $\underline{72.85}_{\pm4.95}$ | $22.91_{\pm2.69}$ |
| DGI(LP) | $62.77_{\pm6.18}$ | $66.80_{\pm6.22}$ | $66.34_{\pm3.55}$ | $60.83_{\pm6.95}$ | $69.90_{\pm10.56}$ | $22.46_{\pm2.83}$ |
| GraphMAE(LP) | $63.02_{\pm5.79}$ | $\underline{67.35}_{\pm6.31}$ | $\underline{66.46}_{\pm2.70}$ | $64.15_{\pm5.14}$ | $72.12_{\pm5.77}$ | $24.41_{\pm2.93}$ |
| GPPT | $60.08_{\pm6.60}$ | $62.98_{\pm7.27}$ | $66.18_{\pm3.76}$ | $65.41_{\pm8.24}$ | $69.47_{\pm9.98}$ | $22.71_{\pm2.93}$ |
| GPF | $62.52_{\pm5.90}$ | $65.77_{\pm6.34}$ | $66.03_{\pm4.91}$ | $61.41_{\pm5.98}$ | $71.90_{\pm6.66}$ | $22.42_{\pm2.68}$ |
| GraphPrompt | $62.67_{\pm6.32}$ | $62.12_{\pm6.88}$ | $63.90_{\pm4.02}$ | $55.97_{\pm5.32}$ | $64.16_{\pm9.48}$ | $23.29_{\pm3.02}$ |
| EdgePrompt | $58.22_{\pm6.71}$ | $63.39_{\pm6.65}$ | $58.01_{\pm8.23}$ | $57.54_{\pm7.60}$ | $71.06_{\pm8.18}$ | $22.24_{\pm2.69}$ |
| DAGPrompT | $63.50_{\pm6.07}$ | $65.47_{\pm5.84}$ | $66.35_{\pm2.77}$ | $52.87_{\pm5.08}$ | $67.64_{\pm9.70}$ | $23.44_{\pm2.69}$ |
| GraphLoRA-S | $60.30_{\pm6.61}$ | $65.46_{\pm6.06}$ | $65.18_{\pm4.09}$ | $\underline{66.73}_{\pm3.96}$ | $67.25_{\pm10.78}$ | $\mathbf{24.50}_{\pm2.74}$ |
| GP2F(Ours) | $\mathbf{66.87}_{\pm5.79}$ | $\mathbf{69.18}_{\pm5.00}$ | $\mathbf{66.49}_{\pm3.37}$ | $\mathbf{66.78}_{\pm3.14}$ | $\mathbf{73.19}_{\pm8.20}$ | $\underline{24.49}_{\pm2.78}$ |

GPF (Fang et al., 2023), GraphPrompt (Liu et al., 2023), EdgePrompt (Fu et al., 2025), and DAGPrompT (Chen et al., 2025). (4) Cross-domain graph prompt learning methods: GraphLoRA (Yang et al., 2025). GraphLoRA-S denotes GraphLoRA without the SMMD loss, because source domain features are unavailable in cross-domain prompt learning. Cross-dataset transfer is treated as a practical form of cross-domain transfer, as methods like GRACE are not designed for multi-domain settings. Since our benchmarks cover citation, co-purchase, co-authorship, and webpage networks, they provide an effective testbed for cross-domain generalization. Detailed experimental settings and dataset statistics are provided in Appendix C.

### 5.2. Performance of Cross-Domain Classification (RQ1)

To evaluate the effectiveness of GP2F, we conduct extensive experiments on seven node classification datasets and six graph classification datasets. For GPL methods, encoder is pre-trained with GRACE. The results are summarized in Table 1, Table 2, and Fig. 3. Overall, GP2F achieves

either the best or second-best performance in most cases, demonstrating its superiority in cross-domain scenarios.

**1-shot Node Classification.** As shown in Table 1, our method outperforms existing GPL methods across most datasets. Specifically, compared to the best baseline, GP2F improves the performance by 1.66% on average. In particular, the improvements on *CiteSeer* and *CS* datasets are more significant, reaching 3.45% and 3.49%, respectively. Another important observation is that Linear Probing (LP) methods in the second block show very stable performance. LP achieves the best result on *PubMed* and is the runner-up on *Computers* and *WikiCS*. It also remains competitive with GPL methods on other datasets. This further supports our view that combining a frozen pre-trained model with an adapted branch can yield excellent results in cross-domain tasks. Meanwhile FT performs worse than LP on all datasets because in 1-shot scenarios, where labels are extremely scarce, full fine-tuning may cause overfitting and forgetting of general knowledge within the pre-trained GNN, leading to performance degradation.

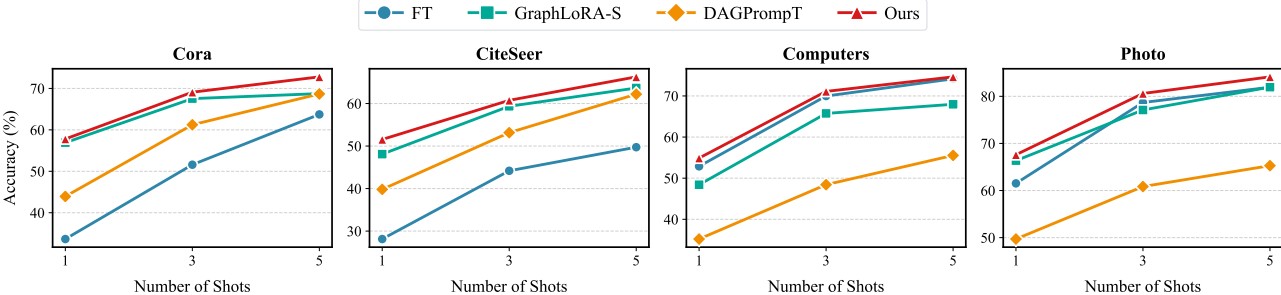

*Figure 3.* Accuracy of 3-shot and 5-shot cross-domain node classification experiments using `GRACE` for pre-training.

*Table 3.* Accuracy of 1-shot cross-domain node classification following the setting of (Wang et al., 2025). Methods marked with * are reported from (Wang et al., 2025). Best in **bold**.

| Method | Cora | CiteSeer | PubMed |
|---|---|---|---|
| GCOPE* | $34.23_{\pm 8.16}$ | $39.05_{\pm 8.82}$ | $44.85_{\pm 6.72}$ |
| MDGPT* | $39.54_{\pm 9.02}$ | $39.24_{\pm 8.95}$ | $45.39_{\pm 11.01}$ |
| MDGFM* | $44.83_{\pm 7.41}$ | $42.18_{\pm 6.41}$ | $46.84_{\pm 7.31}$ |
| GP2F(Ours) | $\mathbf{49.27}_{\pm 7.88}$ | $\mathbf{45.02}_{\pm 7.03}$ | $\mathbf{64.62}_{\pm 3.82}$ |

*Table 4.* Accuracy of node classification on large-scale graph datasets."OOM" denotes "Out of Memory". Methods marked with * are reported from (Yang et al., 2025). Best in **bold**.

| Method | ogbn-arxiv | ogbn-products |
|---|---|---|
| # Node numbers | 169,343 | 2,449,029 |
| # Edge numbers | 1,166,243 | 61,859,140 |
| GPPT* | $65.82_{\pm 0.23}$ | $67.93_{\pm 0.27}$ |
| GPF* | $67.11_{\pm 0.17}$ | $74.04_{\pm 0.50}$ |
| GraphPrompt* | $57.62_{\pm 0.08}$ | OOM |
| GraphLoRA* | $68.61_{\pm 0.20}$ | $75.05_{\pm 0.12}$ |
| GP2F(Ours) | $\mathbf{68.76}_{\pm 0.59}$ | $\mathbf{76.91}_{\pm 0.43}$ |

**50-shot Graph Classification.** We further conduct few-shot graph classification, with results presented in Table 2. We report the AUROC (AUC) for binary classification datasets and Accuracy (ACC) for multi-class datasets. `GP2F` achieves the best or second-best performance on all datasets, with the most significant improvements appear on *BZR* (1.83%) and *COX2* (3.02%). Similar to node classification tasks, we observe that LP-based methods are competitive with state-of-the-art GPL methods.

**1/3/5-shot Node Classification.** We also evaluate the performance of `GP2F` in 3-shot and 5-shot scenarios. As illustrated in Fig. 3, our method consistently outperforms baselines. Additionally, we found that `FT` becomes more stable on larger datasets with more labels. For example, `FT` performs well on *Computers* and *Photo*, achieving significant improvement in 5-shot scenario on *Computers*. However, it still struggles on smaller datasets like *Cora* and *CiteSeer*.

### 5.3. Robustness Across Pre-training Strategies (RQ2)

We evaluate the robustness of our method across different pre-training strategies, including `DGI` and `GraphMAE`. Specifically, `DGI` uses a contrastive objective with cross-entropy loss, while `GraphMAE` is a generative approach. The results in Table 5 show that our method achieves the best or second-best performance on most datasets under both pre-training methods. This indicates that `GP2F` is compatible with various pre-trained models, regardless of whether they use contrastive or generative objectives. These results further demonstrate that `GP2F` can effectively utilize knowledge from different types of pre-training paradigms.

### 5.4. Comparison with Graph Foundation Models (RQ3)

We evaluate the performance of our method against existing Graph Foundation Models (GFMs) including `GCOPE` (Zhao et al., 2024), `MDGPT` (Yu et al., 2024b) and `MDGFM` (Wang et al., 2025) by pre-training with `GRACE`. As shown in Table 3, our method consistently achieves the best performance across three datasets. Interestingly, we observe that pre-training on multiple domains sometimes results in lower accuracy than pre-training on a single domain. This may be because massive semantic and topological gaps between different source domains introduce additional noise during the pre-training phase, which ultimately degrades the quality of the pre-trained representations.

### 5.5. Scalability on Large-Scale Datasets (RQ4)

We evaluate the applicability of `GP2F` on large-scale graph datasets by conducting experiments on *ogbn-arxiv* and *ogbn-products* with standard data splits (Hu et al., 2020). We adopt a sampling strategy during training and evaluation due to the large data scale. Specifically, for each node, we sample 20 neighbors at each layer, corresponding to 2-hop neighborhood sampling for our two-layer GCN. The results show that `GP2F` maintains stable performance on these large-scale datasets and consistently outperforms baselines.

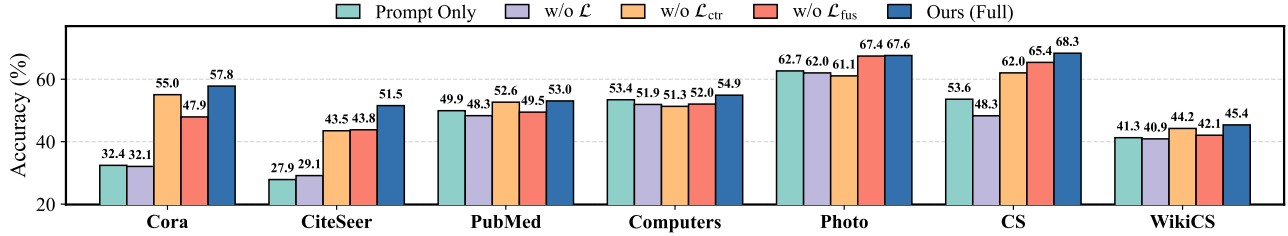

*Figure 4.* Analysis of key components in GP2F via 1-shot node classification pre-trained with GRACE.

*Table 5.* Accuracy of 1-shot node classification with different pre-training methods. Best in **bold**.

| Pre-training + Tuning | Cora | CiteSeer | PubMed | Computers | Photo | CS | WikiCS |
|---|---|---|---|---|---|---|---|
| DGI+FT | $34.17_{\pm10.01}$ | $28.02_{\pm7.70}$ | $51.46_{\pm8.81}$ | $48.40_{\pm12.37}$ | $61.53_{\pm11.28}$ | $54.07_{\pm10.21}$ | $39.75_{\pm9.46}$ |
| DGI+GraphLoRA-S | $55.44_{\pm8.36}$ | $47.85_{\pm10.74}$ | $50.89_{\pm13.15}$ | $48.50_{\pm10.75}$ | $65.03_{\pm10.71}$ | $58.77_{\pm6.46}$ | $45.34_{\pm7.95}$ |
| DGI+DAGPrompT | $49.76_{\pm9.45}$ | $45.73_{\pm10.11}$ | $51.29_{\pm9.81}$ | $30.83_{\pm12.23}$ | $53.59_{\pm8.85}$ | $66.90_{\pm8.02}$ | $28.54_{\pm6.67}$ |
| DGI+GP2F(Ours) | $\mathbf{59.14}_{\pm8.80}$ | $\mathbf{49.22}_{\pm9.73}$ | $\mathbf{51.80}_{\pm9.64}$ | $\mathbf{54.64}_{\pm10.09}$ | $\mathbf{67.29}_{\pm10.60}$ | $\mathbf{69.74}_{\pm8.04}$ | $\mathbf{45.95}_{\pm8.88}$ |
| GraphMAE+FT | $35.12_{\pm10.84}$ | $30.30_{\pm7.52}$ | $52.17_{\pm9.14}$ | $51.29_{\pm12.86}$ | $62.22_{\pm11.39}$ | $57.89_{\pm9.67}$ | $42.76_{\pm9.13}$ |
| GraphMAE+GraphLoRA-S | $56.04_{\pm9.24}$ | $47.01_{\pm9.05}$ | $48.00_{\pm14.48}$ | $50.58_{\pm11.97}$ | $65.32_{\pm11.26}$ | $64.47_{\pm9.07}$ | $45.87_{\pm8.96}$ |
| GraphMAE+DAGPrompT | $55.50_{\pm9.82}$ | $\mathbf{55.63}_{\pm10.42}$ | $\mathbf{52.84}_{\pm9.72}$ | $43.44_{\pm11.72}$ | $56.07_{\pm9.56}$ | $66.46_{\pm7.88}$ | $34.43_{\pm7.36}$ |
| GraphMAE+GP2F(Ours) | $\mathbf{56.08}_{\pm10.27}$ | $51.26_{\pm9.65}$ | $52.18_{\pm10.50}$ | $\mathbf{51.50}_{\pm9.41}$ | $\mathbf{66.59}_{\pm10.40}$ | $\mathbf{67.22}_{\pm7.68}$ | $\mathbf{48.11}_{\pm8.44}$ |

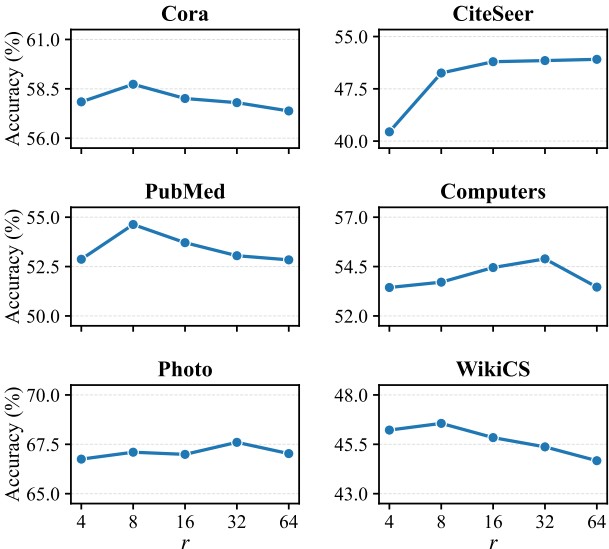

*Figure 5.* Hyperparameter analysis of $r$ for 1-shot node classification with GRACE pre-trained.

### 5.6. Model Analysis (RQ5)

To validate the effectiveness of each component in GP2F, we conduct ablation studies and hyperparameter analysis experiments, as shown in Fig. 4 and Fig. 5.

**Ablation Study.** We design four variants: (1) w/o $\mathcal{L}$, which removes both $\mathcal{L}_{ctr}$ and $\mathcal{L}_{fus}$; (2) w/o $\mathcal{L}_{ctr}$, which removes $\mathcal{L}_{ctr}$; (3) w/o $\mathcal{L}_{fus}$, which removes $\mathcal{L}_{fus}$; and (4) Prompt Only, which keeps only the adapted branch. As shown in

Fig. 4, removing any loss function leads to a performance drop across all datasets, which shows the effectiveness of both $\mathcal{L}_{ctr}$ and $\mathcal{L}_{fus}$. Furthermore, we observe that the variant without any loss (w/o $\mathcal{L}$) performs the worst in most cases. This proves that directly merging the representations of the two branches without constraints leads to a decrease in performance, due to the semantic conflicts between them.

**Hyperparameter Analysis.** To examine the sensitivity of GP2F to key hyperparameters, we analyze the adapter projection dimension $r$, as is shown in Fig. 5. The optimal $r$ varies across datasets. In particular, smaller datasets such as *Cora* and *CiteSeer* achieve better performance with smaller $r$, while larger datasets such as *Computers* and *Photo* favor larger $r$. Since $r$ determines the learning capacity of the adapters, excessively large $r$ may lead to overfitting, whereas overly small $r$ may result in underfitting.

## 6. Conclusion

In this work, we investigate the factors that contribute to the effectiveness of existing Graph Prompt Learning (GPL) methods in cross-domain settings. We reveal that LP and FT serve as strong baselines in cross-domain scenarios through experiment and jointly optimizing a frozen branch and an adapted branch yields a smaller evaluation error than relying on either branch alone through theoretical analysis. Guided by this finding, we propose GP2F, a novel cross-domain graph prompt learning method. GP2F integrates a frozen pre-trained GNN branch and an adapter-based adapted branch, and employs a contrastive alignment loss $\mathcal{L}_{ctr}$ together with a topology-consistent fusion loss $\mathcal{L}_{fus}$

to jointly regulate cross-branch alignment and fusion. Extensive experiments demonstrate that GP2F consistently outperforms existing methods in cross-domain few-shot settings. Overall, this work reveals the underlying factors of the effectiveness of GPL in cross-domain settings and propose a novel GPL method for cross-domain scenarios.

## Impact Statement

This paper presents work whose goal is to advance the field of Machine Learning. There are many potential societal consequences of our work, none which we feel must be specifically highlighted here.

## Acknowledgments

This work was supported by the National Natural Science Foundation of China (No. 62422210, No. 62276187, No.92370111, and No. 62272340)

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

# A. Theorems

## A.1. Proof for Lemma 3.3

*Proof.* From Eq. (9),

$$\text{MSE}(\lambda) = \mathbb{E}\big[\|\lambda\boldsymbol{\epsilon}_i^g + (1-\lambda)\boldsymbol{\epsilon}_i^a\|^2\big] = \lambda^2\sigma_g^2 + (1-\lambda)^2\sigma_a^2 + 2\lambda(1-\lambda)\rho. \tag{25}$$

Expanding $(1-\lambda)^2$ and collecting terms yields $\text{MSE}(\lambda) = A\lambda^2 + B\lambda + C$ with $A = \sigma_g^2 + \sigma_a^2 - 2\rho$, $B = 2(\rho - \sigma_a^2)$, and $C = \sigma_a^2$. Since $A > 0$ by Assumption 3.2, $\text{MSE}(\lambda)$ is strictly convex and thus has a unique minimizer $\lambda^\star = -B/(2A) = (\sigma_a^2 - \rho)/(\sigma_g^2 + \sigma_a^2 - 2\rho)$. Moreover, $\rho < \sigma_a^2$ implies $\lambda^\star > 0$ and $\rho < \sigma_g^2$ implies $\lambda^\star < 1$. Then we substitute $\lambda^\star$ to give the minimum value:

$$\text{MSE}(\lambda^\star) = C - \frac{B^2}{4A} = \sigma_a^2 - \frac{(\sigma_a^2 - \rho)^2}{\sigma_g^2 + \sigma_a^2 - 2\rho} = \sigma_g^2 - \frac{(\sigma_g^2 - \rho)^2}{\sigma_g^2 + \sigma_a^2 - 2\rho}. \tag{26}$$

Under Assumption 3.2, we have $A > 0$, $\sigma_a^2 - \rho > 0$, and $\sigma_g^2 - \rho > 0$. Therefore, the fractions in Eq. (26) are strictly positive, which implies $\text{MSE}(\lambda^\star) < \min\{\sigma_g^2, \sigma_a^2\}$. $\square$

## A.2. Proof for Theorem 3.4

*Proof.* By Lemma 3.3, $\text{MSE}(\lambda)$ is strictly convex and attains its unique minimum at $\lambda^\star \in (0,1)$. Hence $\text{MSE}(\lambda^\star) < \text{MSE}(0)$ and $\text{MSE}(\lambda^\star) < \text{MSE}(1)$. Noting that $\text{MSE}(0) = \mathbb{E}\|\mathbf{h}_i^a - \mathbf{z}_i\|^2$ and $\text{MSE}(1) = \mathbb{E}\|\mathbf{h}_i^g - \mathbf{z}_i\|^2$ completes the proof. $\square$

To connect representation estimation to classification, we show that under Assumption 3.1, a smaller $\mathbb{E}\|\tilde{\mathbf{z}}_i(\lambda) - \mathbf{z}_i\|^2$ yields a tighter upper bound on the misclassification probability.

## A.3. Assumption A.1

**Assumption A.1** (Bounded classifier norm). *There exists a constant $B > 0$ such that $\|\mathbf{w}_c^\star\| \le B$ for all $c \in \{1, \ldots, C\}$.*

## A.4. Corollary A.2

**Corollary A.2** (Margin-based misclassification bound). *Assume Assumptions 3.1, A.1, and 3.2. Let $\hat{y}_i(\lambda) := \arg\max_c (\mathbf{w}_c^\star)^\top \tilde{\mathbf{z}}_i(\lambda)$. Then for any $\lambda$,*

$$\mathbb{P}\big(\hat{y}_i(\lambda) \ne y_i\big) \le \frac{4(C-1)B^2}{\gamma^2} \cdot \mathbb{E}\big[\|\tilde{\mathbf{z}}_i(\lambda) - \mathbf{z}_i\|^2\big].$$

*In particular, Theorem 3.4 implies a strictly tighter upper bound at $\lambda^\star$.*

*Proof.* For any $c \ne y_i$, let $\mathbf{u}_{i,c} := \mathbf{w}_{y_i}^\star - \mathbf{w}_c^\star$. By Assumption 3.1, $\mathbf{u}_{i,c}^\top \mathbf{z}_i \ge \gamma$. Let $\boldsymbol{\delta}_i(\lambda) := \tilde{\mathbf{z}}_i(\lambda) - \mathbf{z}_i$. If $\hat{y}_i(\lambda) \ne y_i$, then for some $c \ne y_i$, $\mathbf{u}_{i,c}^\top \tilde{\mathbf{z}}_i(\lambda) \le 0$, which implies $|\mathbf{u}_{i,c}^\top \boldsymbol{\delta}_i(\lambda)| \ge \gamma$. By the union bound and Markov's inequality,

$$\mathbb{P}\big(\hat{y}_i(\lambda) \ne y_i\big) \le \sum_{c \ne y_i} \frac{\mathbb{E}\big[(\mathbf{u}_{i,c}^\top \boldsymbol{\delta}_i(\lambda))^2\big]}{\gamma^2}.$$

Using $(\mathbf{u}^\top \boldsymbol{\delta})^2 \le \|\mathbf{u}\|^2 \|\boldsymbol{\delta}\|^2$ and $\|\mathbf{u}_{i,c}\| \le \|\mathbf{w}_{y_i}^\star\| + \|\mathbf{w}_c^\star\| \le 2B$ yields the claim. $\square$

# B. Algorithm

---

**Algorithm 1** Training Procedure of `GP2F`

---

**Input:** Graph $\mathbf{G} = (\mathbf{A}, \mathbf{X})$ with labels $y$; Pre-trained GNN $g_{\theta*}$; Adapters $\mathcal{A}$; Projector $Proj(\cdot)$; Linear classifier $\mathbf{C}(\cdot)$; Temperature parameters $\tau_{\text{ctr}}$ and $\tau_{\text{fus}}$; Loss weights $\lambda_1$ and $\lambda_2$; Learnable scaling factors $\alpha$ and $\beta = \{\beta^{(l)}\}_{l=1}^{L}$.
**Output:** $\mathcal{A}$; $Proj(\cdot)$; $\mathbf{C}(\cdot)$; scaling factors $\alpha$ and $\beta$.
**Initialization:** Parameters of $\mathcal{A}$, $Proj(\cdot)$ and $\mathbf{C}(\cdot)$; $\alpha$ and $\beta$; freeze $g_{\theta*}$.
**while** not converged **do**
    # Dimension alignment
    Project node features: $\mathbf{H}^{(0)} \leftarrow Proj(\mathbf{X})$;
    # Dual-branch encoding
    Frozen-branch encoding: $\mathbf{H}_{\text{pre}} \leftarrow g_{\theta*}(\mathbf{A}, \mathbf{H}^{(0)})$;                            (Eq. (13))
    Adapter-branch encoding with layer-wise adapters and scaling factors: $\mathbf{H}_{\text{adp}} \leftarrow g_{\theta*}(\mathbf{A}, \mathbf{H}^{(0)}, \mathcal{A}, \beta)$;     (Eq. (16))
    # Cross-branch contrastive alignment
    Construct positive pairs: $\mathcal{P}(\mathbf{A})$;                                   (Eq. (17))
    Compute contrastive loss: $\mathcal{L}_{\text{ctr}}(\mathbf{H}_{\text{pre}}, \mathbf{H}_{\text{adp}}, \mathcal{P})$;                        (Eqs. (18), (19))
    # Representation fusion
    Fuse branch representations: $\mathbf{H}_{\text{mix}} = \alpha \cdot \mathbf{H}_{\text{pre}} + (1 - \alpha) \cdot \mathbf{H}_{\text{adp}}$;              (Eq. (15))
    # Topology-consistent regularization
    Compute normalized self-similarity matrices: $\mathbf{S}_{\text{pre}} = \text{Norm}(\mathbf{H}_{\text{pre}}\mathbf{H}_{\text{pre}}^{\top})$, $\mathbf{S}_{\text{adp}} = \text{Norm}(\mathbf{H}_{\text{adp}}\mathbf{H}_{\text{adp}}^{\top})$;    (Eq. (20))
    Compute fused similarity matrix: $\mathbf{S}_{\text{mix}} = \alpha \cdot \mathbf{S}_{\text{pre}} + (1 - \alpha) \cdot \mathbf{S}_{\text{adp}}$;            (Eq. (21))
    Construct consistency mask: $\mathcal{M}(\mathbf{A}, \mathbf{S}_{\text{mix}})$;                            (Eq. (22))
    Compute fusion loss: $\mathcal{L}_{\text{fus}}(\mathbf{S}_{\text{mix}}, \mathbf{A}, \mathcal{M})$;                             (Eq. (23))
    # Downstream classification
    Classification loss: $\mathcal{L}_{\text{cls}}(\mathbf{C}(\mathbf{H}_{\text{mix}}), y)$;
    # Overall objective and optimization
    Total loss: $\mathcal{L} \leftarrow \mathcal{L}_{\text{cls}} + \lambda_1 \mathcal{L}_{\text{ctr}} + \lambda_2 \mathcal{L}_{\text{fus}}$;                            (Eq. (24))
    Update parameters of $\mathcal{A}$, $Proj(\cdot)$ and $\mathbf{C}(\cdot)$, $\alpha$ and $\beta$ by minimizing $\mathcal{L}$.
**end while**
**return** $\mathcal{A}$, $Proj(\cdot)$, $\mathbf{C}(\cdot)$, $\alpha$, and $\beta$.

---

# C. Experiment setting and datasets

## C.1. Datasets

We introduce the details of the 15 commonly used real-world datasets, 9 for node classification as is shown in Table 6, and 6 for graph classification as is shown in Table 7.

- *Cora*, *CiteSeer*, *PubMed* (Yang et al., 2016), and *ogbn-arxiv* (Hu et al., 2020) are citation networks where nodes represent scientific papers and edges denote citation relationships. Node features correspond to bag-of-words representations or word embeddings of the paper content. Labels indicate the academic topic of the paper.

- *Amazon Computers*, *Amazon Photo* (Shchur et al., 2018), and *ogbn-products* (Hu et al., 2020) are co-purchase networks collected from Amazon. Nodes represent goods, and edges indicate that connected products are frequently bought together. Node features are derived from product reviews or descriptions.

- *Coauthor CS* (Shchur et al., 2018) is a co-authorship network where nodes represent authors and edges represent scientific collaborations. Features are bag-of-words vectors of keywords from the authors' papers.

- *WikiCS* (Mernyei & Cangea, 2020) is a web-link graph derived from Wikipedia. Nodes correspond to computer science articles, and edges represent hyperlinks between them. Node features are averaged GloVe word embeddings of the article text.

- *MUTAG* (Debnath et al., 1991), *COX2*, and *BZR* (Morris et al., 2020) are molecular graph datasets, where nodes

*Table 6.* Statistics of node classification datasets.

| Dataset | #Nodes | #Edges | #Features | #Classes |
|---|---|---|---|---|
| Cora | 2,708 | 5,429 | 1,433 | 7 |
| CiteSeer | 3,327 | 4,732 | 3,703 | 6 |
| PubMed | 19,717 | 44,338 | 500 | 3 |
| Computers | 13,752 | 245,861 | 767 | 10 |
| Photo | 7,650 | 119,081 | 745 | 8 |
| CS | 18,333 | 81,894 | 6,805 | 15 |
| WikiCS | 11,701 | 216,123 | 300 | 10 |
| ogbn-arxiv | 169,343 | 1,166,243 | 128 | 40 |
| ogbn-products | 2,449,029 | 61,859,140 | 100 | 47 |

*Table 7.* Statistics of graph classification datasets.

| Dataset | #Graphs | #Nodes | #Edges | #Features | #Classes |
|---|---|---|---|---|---|
| COX2 | 467 | 41.2 | 43.4 | 35 | 2 |
| BZR | 405 | 35.8 | 38.4 | 53 | 2 |
| DD | 1,178 | 284.32 | 715.7 | 89 | 2 |
| PROTEINS | 1,113 | 39.1 | 72.8 | 3 | 2 |
| MUTAG | 188 | 17.9 | 19.8 | 7 | 2 |
| ENZYMES | 600 | 32.6 | 62.1 | 3 | 6 |

denote atoms and edges represent chemical bonds. These datasets are commonly used for graph-level classification of molecular properties, such as mutagenicity or biological activity.

- *PROTEINS* (Dobson & Doig, 2003), *ENZYMES* (Borgwardt et al., 2005), and *DD* (Shervashidze et al., 2011) are protein structure datasets from bioinformatics, in which nodes correspond to secondary structure elements (SSEs) and edges encode spatial or sequential proximity between them. The task is to perform graph-level protein classification.

*Table 8.* Settings and code links of various baseline methods.

| Methods | Source Code |
|---|---|
| *k*-Shot Sampling | ProG/blob/main/prompt_graph/tasker/node_task.py |
| Dataset Split | ProG/blob/main/prompt_graph/data/load4data.py |
| Evaluation | ProG/blob/main/prompt_graph/evaluation/AllInOneEva.py |
| DGI | https://github.com/PetarV-/DGI |
| GRACE | https://github.com/CRIPAC-DIG/GRACE |
| GraphMAE | https://github.com/THUDM/GraphMAE/tree/pyg |
| GPPT | https://github.com/MingChen-Sun/GPPT |
| GraphPrompt | https://github.com/Starlien95/GraphPrompt |
| GPF | https://github.com/zjunet/GPF |
| EdgePrompt | https://github.com/xbfu/EdgePrompt |
| DAGPrompT | https://github.com/Cqkkkkkk/DAGPrompT |
| GraphLoRA | https://github.com/AllminerLab/GraphLoRA |

### C.2. Baselines

**Graph Pre-training Methods.**

- DGI (Veličković et al., 2019) is a contrastive pre-training method that uses the original graph and a corrupted version as two views. The goal is to minimize the mutual information between node embeddings and the global graph representation, enabling the model to learn meaningful node representations.

- GRACE (Zhu et al., 2020) constructs two augmented views of the same graph through operations such as edge dropping and feature masking, and trains the model to bring representations of the same node closer across views while separating representations of different nodes using a InfoNCE loss.

- GraphMAE (Hou et al., 2022) randomly masks node features and learns node representations by reconstructing the masked attributes with an encoder–decoder framework, which encourages the model to capture informative structural and semantic patterns.

**Graph Prompt Learning Methods.**

- GPPT (Sun et al., 2022) is the first work that introduces the "pre-training–prompting" paradigm into graph learning. It pre-trains GNNs using link prediction and designs task tokens and structure tokens for downstream adaptation. The task tokens serve as learnable class prototypes obtained via clustering, while the structure tokens aggregate neighborhood information of target nodes. Downstream node classification is reformulated as predicting whether a link exists between the two types of tokens.

- GPF (Fang et al., 2023) proposes a universal graph prompt learning framework that can be applied to different pre-training strategies. They inject learnable prompt vector shared across all nodes directly into input node features and feed the prompted nodes into a frozen pre-trained GNN for downstream tasks.

- GraphPrompt (Liu et al., 2023) proposes a general framework that connects graph pre-training with downstream tasks. It reformulates both pre-training and downstream objectives as subgraph similarity-based link prediction problems. For downstream adaptation, GraphPrompt introduces subgraph-level prompt vectors that guide the Readout operation toward task-relevant information, enabling efficient adaptation while keeping the pre-trained GNN fixed.

- EdgePrompt (Fu et al., 2025) proposes a prompt-tuning strategy that adapts pre-trained GNNs by modifying graph structure rather than node features. It injects learnable edge-level prompts into the adjacency matrix to adjust message passing between connected nodes in the hidden layers, enabling adaptation based on graph structure.

- DAGPrompT (Chen et al., 2025) uses subgraph similarity-based link prediction for pre-training. For downstream adaptation, it applies LoRA to the GNN weights $W$ and the message passing process, enabling parameter-efficient fine-tuning of the pre-trained model. It performs layer-wise prediction through the similarity between subgraph embeddings and class prototypes, where a set of learnable class tokens is introduced for task adaptation.

- GraphLoRA (Yang et al., 2025) uses a weighted MMD loss for feature distribution alignment and performs LoRA-based fine-tuning of the GNN weights $W$ under a contrastive constraint for topological alignment, together with an additional structure-aware regularization loss for classification.

**Graph Foundation Models.**

- GCOPE (Zhao et al., 2024) introduces a cross-domain graph pre-training framework that connects multiple source graphs through learnable coordinators. These coordinators link isolated datasets into a unified graph for joint pre-training, enabling the model to learn transferable representations across domains while preserving domain-specific characteristics. The learned representations can be adapted via fine-tuning or graph prompting in a parameter-efficient manner.

- MDGPT (Yu et al., 2024b) proposes a dual-prompt design for downstream adaptation, consisting of a unifying prompt and a mixing prompt. The mixing prompt aligns the target domain with cross-domain knowledge learned during pre-training, while the unifying prompt supports finer-grained domain-specific adaptation through learnable prompt vectors.

- MDGFM (Wang et al., 2025) performs multi-domain graph pre-training by jointly leveraging multiple source graphs with contrastive objectives. It further aligns source graph topologies into a shared semantic space using topology-aware refinement via domain tokens. During downstream adaptation, meta-prompts generated by mixing domain tokens capture global transferable knowledge and task-specific prompts enable domain adaptation.

*Table 9.* Hyperparameter settings of `GP2F` in 1-shot scenario.

| Dataset | $up\_lr$ | $down\_lr$ | $up\_wd$ | $down\_wd$ | $\tau_{\text{ctr}}$ | $\tau_{\text{fus}}$ |
|---|---|---|---|---|---|---|
| Cora | 0.0005 | 0.001 | 0.0005 | 0.0005 | 0.5 | 0.05 |
| CiteSeer | 0.0001 | 0.005 | 0.0005 | 0.0005 | 0.2 | 0.1 |
| PubMed | 0.00005 | 0.01 | 0.0005 | 0.0005 | 0.2 | 0.1 |
| Computers | 0.00005 | 0.0001 | 0.0005 | 0.0005 | 0.5 | 0.05 |
| Photo | 0.00005 | 0.001 | 0.0005 | 0.0005 | 0.5 | 0.05 |
| CS | 0.0001 | 0.0001 | 0.0005 | 0.0005 | 0.5 | 0.05 |
| WikiCS | 0.00005 | 0.001 | 0.0005 | 0.0005 | 0.5 | 0.05 |
| COX2 | 0.001 | 0.01 | 0.00001 | 0.0005 | 0.5 | 0.05 |
| BZR | 0.00005 | 0.05 | 0.00001 | 0.0005 | 0.2 | 0.2 |
| DD | 0.0001 | 0.05 | 0.0001 | 0.001 | 0.2 | 0.2 |
| PROTEINS | 0.0005 | 0.01 | 0.00001 | 0.0005 | 0.5 | 0.2 |
| MUTAG | 0.005 | 0.0005 | 0.0001 | 0.00 | 0.2 | 0.05 |
| ENZYMES | 0.0001 | 0.005 | 0.0001 | 0.0005 | 0.2 | 0.05 |

## C.3. Experiment settings

All experiments are conducted on a single NVIDIA GeForce RTX 5090 GPU with 32 GB of memory. For a fair comparison, all other baselines except `GraphLoRA` are implemented with the same 2-layer GCN as ours, while `GraphLoRA` uses its original 2-layer GAT. The pre-training epoch is 1000 while downstream training epoch is 500 with early stopping (patience=20). We tune all training hyperparameters of baselines, and keep all other hyperparameters as reported or suggested in the original papers. Our classifier is a simple linear layer. For each dataset, 90% of the nodes are used for testing, and an N-way K-shot training set is sampled from the remaining 10%. We report the mean and standard deviation over 5 random seeds, each with 100 independent samplings. The seed list is [12345, 23456, 34567, 45678, 56789]. For all methods, the GNN's hidden dimensions are fixed at 128. The learning rates are tuned within [1e-5, 1e-1] and weight decay tuned within [0, 5e-3]. For `GP2F`, we set $r = 32$, and $\lambda_1$ and $\lambda_2$ are tuned within [0.05, 5] while $\tau_{\text{ctr}}$ and $\tau_{\text{fus}}$ is tuned within [0.05, 0.5]. The Adam optimizer is used for optimization. Detailed hyperparameters are shown in Table 9, where $\tau_{\text{ctr}}$ and $\tau_{\text{fus}}$ denotes the hyperparameter in the contrastive loss and fusion loss, respectively.

## D. Related Work

### D.1. Graph Pre-training

Graph pre-training aims to learn transferable representations from unlabeled graphs using self-supervised objectives (Zhuo et al., 2024c;b;a). Existing methods are usually divided into contrastive-based and generative-based approaches. Contrastive methods construct different views of graphs or nodes and encourage consistent representations across views. For example, DGI (Veličković et al., 2019) contrasts node representations with a global graph summary, while GRACE (Zhu et al., 2020) performs node-level contrast across augmented views using an InfoNCE loss. Some work also developed to improve augmentations, such as GraphCL (You et al., 2020), as well as improve stability like SimGRACE (Xia et al., 2022). Generative methods focus on reconstructing corrupted graphs. GraphMAE (Hou et al., 2022) is a masked autoencoder for recovering masked node features with an encoder–decoder architecture, and GraphMAE2 (Hou et al., 2023) further improves reconstruction by introducing multi-view re-masking strategy. In addition, BGRL (Thakoor et al., 2021) follows a bootstrap-style design that learns node representations by predicting a target encoder, avoiding the use of explicit negative samples.

### D.2. Graph Prompt Learning

Graph prompt learning aims to adapt pre-trained GNNs to downstream tasks with lightweight prompts, while keeping the pre-trained encoder largely frozen. This paradigm is motivated by the discrepancy between pre-training and downstream objectives: representations learned from self-supervised tasks may not be directly aligned with downstream prediction tasks. To reduce this objective gap, early methods reformulate downstream tasks into forms that are closer to pre-training objectives.

*Figure 6.* Accuracy of 3-shot and 5-shot cross-domain node classification experiments using `GRACE` for pre-training.

GPPT (Sun et al., 2022) introduces task and structure tokens, and casts node classification as link prediction between nodes and class tokens. Following this reformulation perspective, GraphPrompt (Liu et al., 2023) unifies pre-training and downstream tasks as subgraph similarity learning, while DAGPrompT (Chen et al., 2025) performs layer-wise prediction by measuring similarities between subgraph representations and learnable class prototypes. These methods effectively improve task alignment, but their designs are often tied to specific objective reformulations and may become less stable when the pre-training strategy changes. To improve applicability across different pre-training and downstream tasks, another line of work develops more general prompt mechanisms. Instead of redesigning the downstream objective, these methods modify the input graph or its representations through learnable prompts. GPF (Fang et al., 2023) adds prompt vectors to node features, EdgePrompt (Fu et al., 2025) injects prompts into graph edges to modulate message passing, and All-in-one (Sun et al., 2023) inserts prompt graphs into the original graph to support multiple downstream tasks with improved interpretability. MultiGPrompt (Yu et al., 2024c) further incorporates multi-task pre-training to enhance prompt generality. Although these methods make graph prompting more flexible, they still mainly focus on improving a single adaptation pathway based on the frozen pre-trained encoder. While these studies improve the flexibility of graph prompting, most of them adapt the pre-trained encoder through a single prompting or tuning pathway, and consequently struggles in cross-domain settings.

### D.3. Cross-Domain Graph Prompt Learning and Graph Foundation Models

Recent studies further extend graph prompt learning to cross-domain scenarios, where the source and target graphs may differ in feature distributions, structural patterns, and label semantics. Under such domain shifts, the model needs to preserve useful knowledge from the pre-trained encoder while adapting to the target graph. GraphControl (Zhu et al., 2024) addresses this problem through a control-style adaptation mechanism, which converts downstream features into the GCC input format and injects conditional feature information into frozen structural representations. Despite its effectiveness, it relies on GCC pre-training. GraphLoRA (Yang et al., 2025) instead adds a learnable LoRA-style GNN to frozen representations for structural adaptation, and uses SMMD to align source and target feature distributions. While this improves cross-domain transfer, it requires access to source-domain data during adaptation, which may not be available in practical downstream scenarios. Beyond downstream prompt adaptation, graph foundation models approach cross-domain generalization from the pre-training side. These methods usually aggregate knowledge from multiple source domains and then transfer the learned knowledge to downstream tasks. MDGPT (Yu et al., 2024b) combines shared and domain-specific prompts, MDGFM (Wang et al., 2025) aligns heterogeneous graph structures with topology-aware prompts, GCOPE (Zhao et al., 2024) connects multiple source graphs via learnable coordinators, and BRIDGE (Yuan et al., 2025) learns domain-invariant representations with expert-style routing. In parallel, LLM-enhanced graph foundation models, such as ZeroG (Li et al., 2024), OFA (Liu et al., 2024), GraphCLIP (Zhu et al., 2025), and GraphGPT (Tang et al., 2024), exploit textualization, prompting, or lightweight adaptation to improve graph understanding across tasks and domains. These graph foundation models mainly investigate knowledge transfer across domains, where source-domain knowledge is coordinated during pre-training and transferred to downstream tasks. In contrast, the fusion between general pre-trained knowledge and task-adaptive knowledge is often implemented by simple additive operations, with limited explicit constraint on how the two types of knowledge should be balanced.

## E. Additional Experiments

### E.1. Performance on 1/3/5-shot Node Classification

We report the accuracy of 3-shot and 5-shot scenarios on the remaining datasets, as is shown in Fig. 6. Our method consistently outperforms existing approaches across these datasets. In addition, we observe a notable performance

*Table 10.* Accuracy of 1-shot node classification with different pre-training datasets. Best in **bold**.

| Method | Cora | CiteSeer | PubMed | Computers | Photo | CS | WikiCS |
|---|---|---|---|---|---|---|---|
| | | | Pre-trained on PubMed | | | | |
| GRACE(LP) | $34.85_{\pm9.87}$ | $28.13_{\pm6.72}$ | $50.84_{\pm8.96}$ | $52.13_{\pm12.92}$ | $62.50_{\pm12.16}$ | $58.58_{\pm10.50}$ | $42.06_{\pm8.51}$ |
| GRACE(FT) | $34.82_{\pm9.56}$ | $25.95_{\pm7.53}$ | $51.76_{\pm8.77}$ | $52.78_{\pm11.93}$ | $61.60_{\pm10.94}$ | $55.14_{\pm10.68}$ | $39.83_{\pm8.80}$ |
| GPF | $33.44_{\pm9.29}$ | $28.20_{\pm8.30}$ | $51.66_{\pm8.55}$ | $53.18_{\pm12.20}$ | $62.39_{\pm11.96}$ | $57.48_{\pm10.70}$ | $41.98_{\pm8.80}$ |
| EdgePrompt | $33.03_{\pm8.52}$ | $28.06_{\pm7.50}$ | $48.21_{\pm7.86}$ | $\mathbf{53.46}_{\pm12.17}$ | $63.08_{\pm11.90}$ | $53.60_{\pm10.17}$ | $41.17_{\pm9.57}$ |
| GPPT | $25.49_{\pm9.73}$ | $22.67_{\pm6.26}$ | $44.81_{\pm10.59}$ | $32.10_{\pm14.29}$ | $42.66_{\pm16.59}$ | $33.61_{\pm14.36}$ | $31.61_{\pm10.31}$ |
| GraphPrompt | $48.92_{\pm9.57}$ | $38.75_{\pm8.35}$ | $52.88_{\pm9.42}$ | $42.36_{\pm11.93}$ | $59.30_{\pm9.56}$ | $65.04_{\pm3.97}$ | $35.27_{\pm7.80}$ |
| DAGPrompT | $48.47_{\pm8.85}$ | $40.58_{\pm8.61}$ | $50.90_{\pm9.61}$ | $37.64_{\pm10.95}$ | $51.96_{\pm8.36}$ | $\mathbf{65.88}_{\pm10.01}$ | $30.05_{\pm7.10}$ |
| GraphLoRA-S | $50.11_{\pm10.81}$ | $37.95_{\pm10.92}$ | $50.32_{\pm9.36}$ | $51.64_{\pm11.78}$ | $64.59_{\pm11.46}$ | $43.74_{\pm12.76}$ | $38.91_{\pm8.55}$ |
| GP2F(Ours) | $\mathbf{55.41}_{\pm8.73}$ | $\mathbf{48.93}_{\pm11.42}$ | $\mathbf{53.16}_{\pm10.00}$ | $52.79_{\pm10.28}$ | $\mathbf{66.96}_{\pm10.33}$ | $64.47_{\pm8.58}$ | $\mathbf{45.93}_{\pm8.71}$ |
| | | | Pre-trained on Computers | | | | |
| GRACE(LP) | $34.65_{\pm9.58}$ | $29.01_{\pm6.42}$ | $51.05_{\pm8.61}$ | $53.22_{\pm12.26}$ | $60.75_{\pm10.89}$ | $59.30_{\pm10.41}$ | $42.09_{\pm9.12}$ |
| GRACE(FT) | $33.48_{\pm10.03}$ | $28.63_{\pm7.56}$ | $51.59_{\pm8.86}$ | $52.89_{\pm12.28}$ | $62.15_{\pm10.97}$ | $54.52_{\pm9.50}$ | $39.97_{\pm9.12}$ |
| GPF | $32.99_{\pm10.19}$ | $28.38_{\pm7.53}$ | $51.53_{\pm8.62}$ | $53.17_{\pm12.11}$ | $62.45_{\pm12.01}$ | $57.98_{\pm10.91}$ | $42.26_{\pm9.07}$ |
| EdgePrompt | $30.85_{\pm7.91}$ | $27.09_{\pm6.76}$ | $50.67_{\pm8.12}$ | $53.36_{\pm12.02}$ | $62.12_{\pm11.72}$ | $50.73_{\pm9.68}$ | $40.83_{\pm8.98}$ |
| GPPT | $27.37_{\pm10.81}$ | $24.86_{\pm7.54}$ | $46.02_{\pm10.16}$ | $31.17_{\pm18.39}$ | $39.99_{\pm15.48}$ | $32.49_{\pm14.72}$ | $31.77_{\pm9.29}$ |
| GraphPrompt | $50.85_{\pm10.55}$ | $38.38_{\pm8.40}$ | $52.73_{\pm9.66}$ | $43.78_{\pm11.86}$ | $59.03_{\pm9.46}$ | $\mathbf{70.58}_{\pm7.55}$ | $34.90_{\pm7.85}$ |
| DAGPrompT | $50.46_{\pm9.37}$ | $46.53_{\pm9.27}$ | $51.29_{\pm9.65}$ | $40.62_{\pm11.83}$ | $54.17_{\pm8.75}$ | $68.06_{\pm8.81}$ | $30.45_{\pm6.90}$ |
| GraphLoRA-S | $47.58_{\pm10.81}$ | $33.71_{\pm10.40}$ | $47.94_{\pm12.75}$ | $52.78_{\pm11.74}$ | $65.63_{\pm11.27}$ | $41.47_{\pm12.05}$ | $40.12_{\pm8.37}$ |
| GP2F(Ours) | $\mathbf{59.08}_{\pm9.47}$ | $\mathbf{50.78}_{\pm12.05}$ | $\mathbf{53.64}_{\pm11.56}$ | $\mathbf{54.43}_{\pm10.70}$ | $\mathbf{66.86}_{\pm10.68}$ | $68.48_{\pm8.17}$ | $\mathbf{44.82}_{\pm7.61}$ |

improvement for the compared methods in the 3-shot setting, which may be attributed to the increased number of labeled samples that alleviates overfitting under the extremely label-scarce 1-shot scenario.

### E.2. Hyperparameter Analysis

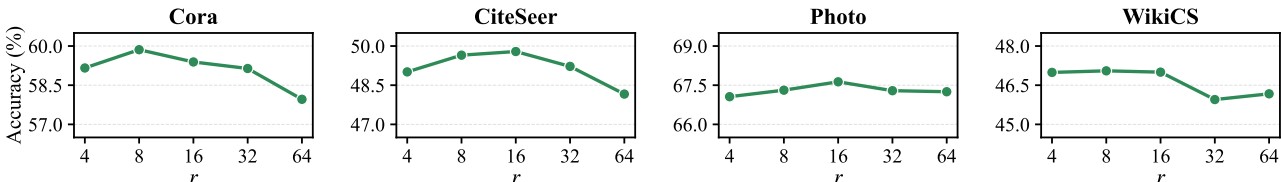

*Figure 7.* Hyperparameter analysis of $r$ for 1-shot node classification with `DGI` pre-trained.

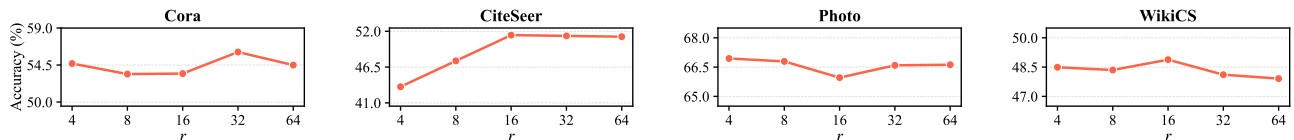

*Figure 8.* Hyperparameter analysis of $r$ for 1-shot node classification with `GraphMAE` pre-trained.

We also conduct a hyperparameter analysis of $r$ with GNNs pre-trained by `DGI` and `GraphMAE`, as shown in Fig. 7 and Fig. 8. Similar to the observations in Fig. 5, the optimal value of $r$ varies across datasets. In practice, $r$ is influenced by both the dataset scale and the degree of cross-domain discrepancy, since it controls the capacity of the adaptation branch to encode task-specific corrections. We generally recommend setting $r$ to 8 or 16 as a starting point, and further tuning it when necessary.

## E.3. Robustness Across Pre-training Datasets

We evaluate the robustness of GP2F across different pre-training datasets. Specifically, we use *PubMed* and *Computers* as source domains, respectively, and report the 1-shot node classification results on diverse downstream datasets. As shown in Table 10, GP2F achieves the best or second-best performance on most datasets under both pre-training settings, demonstrating its effectiveness. Meanwhile, changing the pre-training dataset only leads to limited performance variation, indicating that GP2F is not sensitive to the choice of the pre-training dataset.

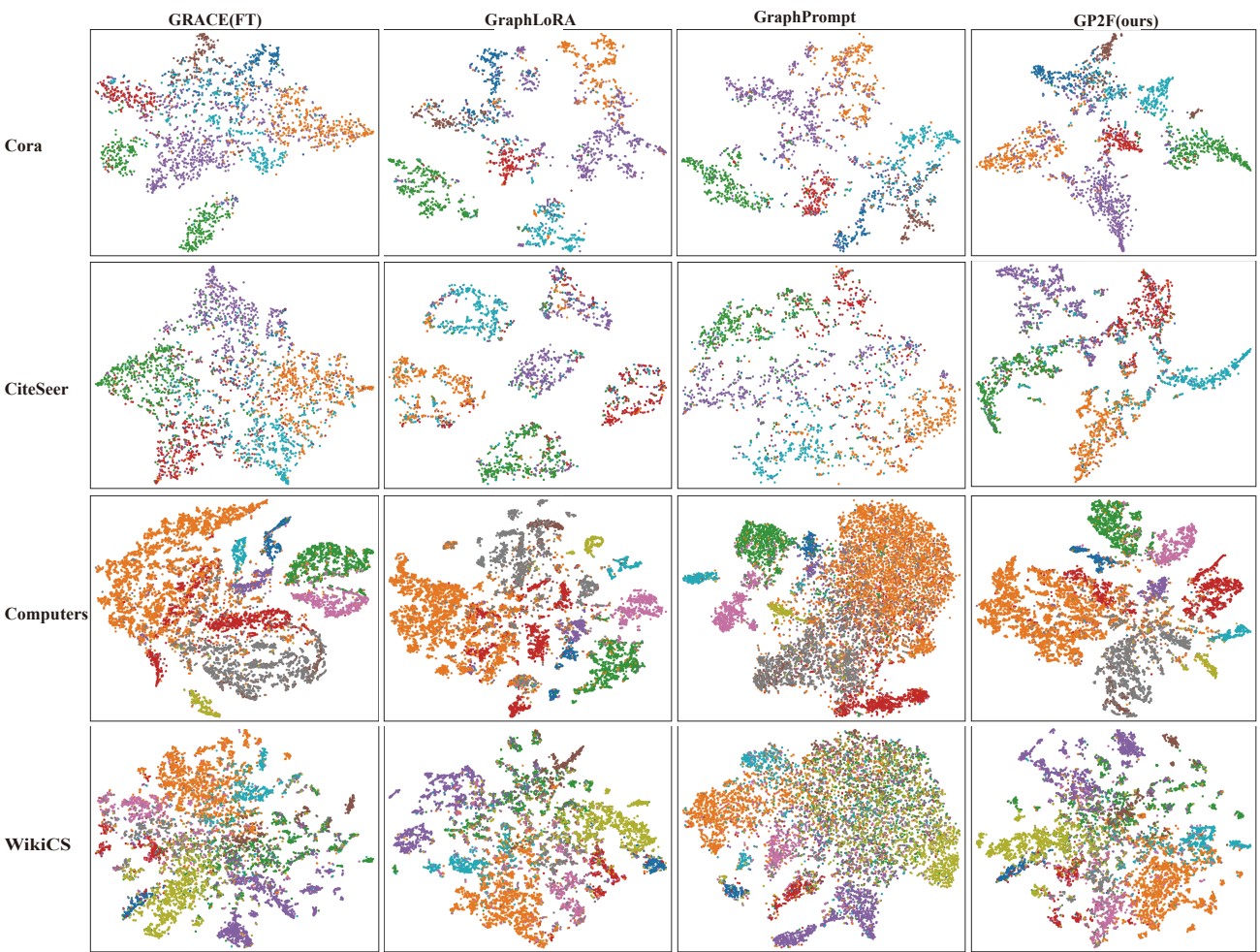

*Figure 9.* Visualization of GP2F and baselines.

## E.4. Efficiency Analysis

*Table 11.* Statistics of trainable parameters, total parameters, and the ratio of trainable parameters to full fine-tuning parameters.

| Metric | GRACE(LP) | GRACE(FT) | GPF | EdgePrompt | GPPT | GraphPrompt | DAGPrompT | GraphLoRA-S | GP2F(Ours) |
|---|---|---|---|---|---|---|---|---|---|
| Trainable Params. (KB) | 1.935 | 202.000 | 3.368 | 3.496 | 30.720 | 0.128 | 16.274 | 60.336 | 18.642 |
| Total Params. (MB) | 0.202 | 0.202 | 0.203 | 0.204 | 0.233 | 0.202 | 0.233 | 0.460 | 0.219 |
| Trainable / FT (%) | 0.96 | 100.00 | 1.67 | 1.73 | 15.21 | 0.06 | 8.06 | 29.87 | 9.23 |

We further report the parameter scale, training cost, and inference cost of different methods in Tables 11, 12, and 13. GP2F introduces 18.642 KB trainable parameters, corresponding to 9.23% of full fine-tuning, indicating its parameter efficiency. In terms of computational cost, GP2F introduces additional overhead mainly due to the auxiliary losses used during training. However, such overhead is still reasonable given the consistent performance improvements obtained across datasets.

*Table 12.* Training time (ms/epoch) and GPU memory (MB) costs across various datasets.

| Methods | Time/Memory | Cora | CiteSeer | PubMed | Computers | Photo | CS | WikiCS |
|---|---|---|---|---|---|---|---|---|
| GRACE(LP) | Time | 2.9 | 3.5 | 3.6 | 6.0 | 3.6 | 14.7 | 5.0 |
| | Memory | 94 | 186 | 334 | 711 | 370 | 942 | 594 |
| GRACE(FT) | Time | 3.2 | 3.9 | 4.4 | 7.1 | 4.9 | 15.2 | 5.1 |
| | Memory | 96 | 188 | 335 | 712 | 372 | 944 | 596 |
| GPF | Time | 3.0 | 3.5 | 3.6 | 6.1 | 4.0 | 14.5 | 5.2 |
| | Memory | 109 | 204 | 442 | 787 | 412 | 1043 | 659 |
| EdgePrompt | Time | 2.5 | 2.8 | 6.8 | 23.1 | 12.3 | 20.5 | 20.6 |
| | Memory | 223 | 288 | 1423 | 5949 | 2914 | 2804 | 5190 |
| GPPT | Time | 6.0 | 5.7 | 4.9 | 10.4 | 7.3 | 21.9 | 9.7 |
| | Memory | 94 | 185 | 334 | 711 | 370 | 941 | 594 |
| GraphPrompt | Time | 5.0 | 4.9 | 4.9 | 6.4 | 5.7 | 8.9 | 5.6 |
| | Memory | 644 | 925 | 2338 | 11003 | 5754 | 13241 | 4161 |
| DAGPrompT | Time | 10.7 | 8.5 | 8.6 | 12.5 | 17.7 | 16.3 | 12.0 |
| | Memory | 399 | 925 | 1014 | 1284 | 715 | 13749 | 461 |
| GraphLoRA-S | Time | 20.1 | 34.3 | 451.7 | 232.3 | 86.2 | 403.7 | 172.5 |
| | Memory | 469 | 718 | 18177 | 10363 | 3615 | 16604 | 7815 |
| GP2F(Ours) | Time | 7.6 | 8.8 | 165.5 | 88.6 | 31.4 | 154.4 | 67.5 |
| | Memory | 533 | 830 | 22739 | 11241 | 3596 | 20237 | 8161 |

*Table 13.* Inference time (ms/epoch) and GPU memory (MB) costs across various datasets.

| Methods | Time/Memory | Cora | CiteSeer | PubMed | Computers | Photo | CS | WikiCS |
|---|---|---|---|---|---|---|---|---|
| GRACE(LP) | Time | 10.8 | 14.9 | 14.0 | 16.4 | 19.4 | 25.7 | 20.6 |
| | Memory | 91 | 167 | 337 | 721 | 374 | 946 | 603 |
| GRACE(FT) | Time | 11.2 | 12.5 | 15.2 | 17.0 | 15.8 | 28.2 | 14.6 |
| | Memory | 109 | 186 | 456 | 805 | 422 | 1057 | 675 |
| GPF | Time | 16.2 | 14.2 | 16.6 | 15.7 | 13.4 | 34.2 | 16.7 |
| | Memory | 106 | 185 | 445 | 796 | 417 | 1046 | 667 |
| EdgePrompt | Time | 18.0 | 17.5 | 18.5 | 40.9 | 27.1 | 40.7 | 44.8 |
| | Memory | 225 | 290 | 1435 | 5960 | 2920 | 2814 | 5199 |
| GPPT | Time | 12.7 | 18.5 | 22.8 | 22.3 | 20.5 | 34.7 | 23.7 |
| | Memory | 91 | 167 | 335 | 721 | 375 | 945 | 604 |
| GraphPrompt | Time | 2543.8 | 3514.5 | 55177.1 | 33950.1 | 13370.4 | 60385.1 | 23829.3 |
| | Memory | 564 | 880 | 1792 | 3961 | 2185 | 13221 | 1416 |
| DAGPrompT | Time | 244.6 | 314.2 | 1718.7 | 1461.7 | 597.8 | 1466.5 | 909.5 |
| | Memory | 440 | 989 | 1044 | 1044 | 753 | 13827 | 493 |
| GraphLoRA-S | Time | 24.7 | 36.3 | 18.6 | 21.6 | 19.6 | 34.8 | 20.7 |
| | Memory | 407 | 631 | 15139 | 7978 | 2695 | 13743 | 5883 |
| GP2F(Ours) | Time | 18.1 | 19.2 | 11.2 | 27.8 | 23.2 | 29.2 | 18.5 |
| | Memory | 483 | 747 | 18857 | 9793 | 3242 | 16930 | 7193 |

### E.5. Visualization

We further provide 5-shot t-SNE visualizations to qualitatively compare the representation distributions learned by `GP2F`, `GRACE(FT)`, `GraphPrompt`, and `GraphLoRA-S`. As shown in Fig. 9, `GP2F` generally produces more compact intra-class clusters and clearer inter-class boundaries across datasets. In contrast, the baseline methods exhibit more mixed class distributions in several cases. These observations suggest that the adaptive fusion of frozen pre-trained knowledge and task-adaptive knowledge helps `GP2F` learn more discriminative target-domain representations.

### E.6. GraphLoRA-S using GCN for fair comparison

*Table 14.* Performance comparison with `GraphLoRA-S` using different backbones. Best in **bold**.

| Method | Cora | CiteSeer | PubMed | Computers | Photo | CS | WikiCS |
|---|---|---|---|---|---|---|---|
| GraphLoRA-S(GCN) | $49.63_{\pm10.92}$ | $39.95_{\pm11.53}$ | $49.38_{\pm13.08}$ | $52.27_{\pm11.94}$ | $65.15_{\pm11.35}$ | $42.47_{\pm14.02}$ | $40.69_{\pm8.58}$ |
| GraphLoRA-S(GAT) | $56.25_{\pm9.13}$ | $48.10_{\pm10.22}$ | $52.68_{\pm12.45}$ | $48.41_{\pm10.03}$ | $66.34_{\pm10.96}$ | $62.72_{\pm8.22}$ | $40.07_{\pm8.97}$ |
| Ours(GCN) | $\mathbf{57.80}_{\pm8.67}$ | $\mathbf{51.55}_{\pm11.73}$ | $\mathbf{53.05}_{\pm10.64}$ | $\mathbf{54.89}_{\pm10.39}$ | $\mathbf{67.60}_{\pm10.93}$ | $\mathbf{68.33}_{\pm7.08}$ | $\mathbf{45.37}_{\pm8.72}$ |

Since the original `GraphLoRA-S` adopts GAT as the backbone, we further evaluate its performance with GCN to ensure a fair comparison with our method. As shown in Table 14, replacing GAT with GCN leads to a clear performance drop for `GraphLoRA-S` on most datasets. This suggests that the reported performance of `GraphLoRA-S` is partly affected by the choice of backbone architecture, and `GP2F` still outperforms `GraphLoRA-S`.

## F. Adaptive Fusion Behavior under Domain Shift

We further discuss how GP2F adapts to domain shift through the learned fusion coefficient $\alpha$. In cross-domain graph prompt learning, source-domain graphs are unavailable during downstream adaptation, making it difficult to explicitly measure the source-target discrepancy or reduce the domain gap through distribution-matching objectives such as MMD. GP2F instead addresses domain shift by balancing the frozen branch, which preserves pre-trained knowledge, and the adapted branch, which captures target-specific information. The coefficient $\alpha$ reflects the relative contribution of the two branches: a larger $\alpha$ indicates greater reliance on the frozen branch, whereas a smaller $\alpha$ assigns more weight to the adapted branch. When the target domain is close to the pre-training domain, $\alpha$ tends to decrease during training, suggesting that the adapted branch gradually becomes reliable. For larger domain shifts, $\alpha$ may slightly increase at the early stage, indicating an initial reliance on the stable frozen branch, and then decreases as the adapted branch learns target-specific knowledge.

## G. Limitations of Theoretical Assumptions

Our theoretical analysis adopts several idealized assumptions for analytical tractability. Specifically, we assume that the latent target representation is linearly separable, and that the frozen and adapted representations are two noisy estimators of the ideal representation $z_i$ with unbiased errors. These assumptions allow us to derive a concise explanation of how error variance and cross-covariance affect the benefit of fusion.

These assumptions do not affect the main conclusion of this paper. Instead, they provide a simplified setting to explain why combining pre-trained knowledge and task-adaptive knowledge can be beneficial. The strong performance of linear probing and fine-tuning in our motivating experiments supports this view: the frozen branch contains useful transferable knowledge, while the adapted branch learns target-specific representations. Therefore, both branches can serve as meaningful estimators of the ideal target representation.

Nevertheless, these assumptions may not hold in extreme cases. If the frozen branch contains little transferable knowledge under severe domain shift, or if the adapted branch overfits the limited few-shot labels, either branch may become a poor estimator of $z_i$. In such cases, the assumptions of unbiased estimation and effective complementarity may be weakened, and the benefit of fusion may diminish. Under the practical settings considered in this paper, however, our empirical results show that the two branches are generally effective and complementary.

