# OpenReview forum: "GP2F: Cross-Domain Graph Prompting with Adaptive Fusion of Pre-trained Graph Neural Networks"
_ICML.cc/2026/Conference — ICML 2026 regular_

### Official Review · Reviewer_tNyP · 2026-03-04

**Soundness:** 3
**Presentation:** 3
**Significance:** 3
**Originality:** 3
**Overall Recommendation:** 5
**Confidence:** 4

**Summary:**

This paper explains and improves the effectiveness of Graph Prompt Learning (GPL) under cross-domain distribution shifts. It proposes GP2F, which employs a frozen branch to preserve pre-trained knowledge and an adaptation branch to inject task-specific signals. The two branches are fused via a topology-aware constraint, with the framework supported by theoretical analysis. Extensive experiments show that the proposed approach outperforms existing baselines.

**Compliance With Llm Reviewing Policy:**

Affirmed.

**Final Justification:**

I hold my prior evaluation and recommend to accept the paper.

**Key Questions For Authors:**

Apart from the weaknesses, please clarify: How is the multi-domain pre-training in Table 3 implemented? What is the relationship between the proposed method and GFMs?

**Limitations:**

yes

**Strengths And Weaknesses:**

Strengths:

1. The paper focuses on a valuable problem. The proposed method addresses the degradation of current GPL baselines in cross-domain settings, which aligns with the frontier of the graph learning.

2. The proposed method is novel and interesting. The well-designed dual-branch architecture disentangles pre-trained knowledge from task-specific adaptation. And the topology-consistent loss makes the fusion mechanism easy to implement.

3. Extensive experiments demonstrate good performance across various settings. Furthermore, the expansion of baselines reflects a high standard of experimental rigor.

Weaknesses:

1. The description of some components should be further clarified. In Line 180, the definition of Proj is not provided; in Line 194, the negative neighbor set is not clearly defined. The sampling strategy should be specified more explicitly.

2. Although the experiments on OGB datasets are impressive, it would be beneficial to include more details on the sampling strategies used during training and inference.

3. GraphLoRA is a strong cross-domain baseline. However, its setting is not consistent with other methods (i.e. GraphLoRA uses GAT while other baselines use GCN). It is unclear whether this bias affects the fairness of cross-domain comparisons.

---

> ### Author Rebuttal · Authors · 2026-03-30
>
> We thank the reviewer for your support of our work and the careful evaluation and constructive remarks. Our detailed responses are provided below.
>
> # W1. Response to concerns on clarity improvement
>
> Thanks. We would like to clarify the experimental setting. Specifically, $Proj(\cdot)$ is a single-layer MLP, which is briefly introduced in Line 234; we will further clarify the definition in the method section for better readability. The negative neighbor set is defined as all samples excluding the positive ones. We will include these definitions and the sampling strategy more explicitly in the final version.
>
> # W2. Response to sampling strategies in OGB experiments
>
> We thank the reviewer for this helpful suggestion. We use mini-batch for training and evaluation, where for each batch we sample a subgraph centered on the nodes in the batch. Specifically, for each node, we sample 20 neighbors at each layer, corresponding to 2-hop neighborhood sampling for our two-layer GCN. We will include this description in the final version for clarity.
>
> # W3. Response to concerns on the fairness of GraphLoRA comparison
>
> Thanks for this suggestion. We have created a table comparing GP2F and GraphLoRA under consistent settings, and the results are shown below:
>
> | Method         | Cora          | CiteSeer        | PubMed        | Computers     | Photo         | CS            | WikiCS       |
> | -------------- | ------------- | --------------- | ------------- | ------------- | ------------- | ------------- | ------------ |
> | GraphLoRA(GCN) | 49.63 ± 10.92 | 39.95 ± 11.53   | 49.38 ± 13.08 | 52.27 ± 11.94 | 65.15 ± 11.35 | 42.47 ± 14.02 | 40.69 ± 8.58 |
> | GraphLoRA(GAT) | 56.25 ± 9.13  | 48.10  ±10.22 | 52.68 ± 12.45 | 48.41 ± 10.03 | 66.34 ± 10.96 | 62.72 ± 8.22  | 40.07 ± 8.97 |
> | Ours(GCN)      | 57.80 ± 8.67  | 51.55 ± 11.73   | 53.05 ± 10.64 | 54.89 ± 10.39 | 67.60 ± 10.93 | 68.33 ± 7.08  | 45.37 ± 8.72 |
>
> Compared to using GAT as the backbone, we observe a performance drop across datasets and our approach still consistently outperforms the baselines. These results will be included in the final version to ensure fair cross-domain comparisons.
>
> # Q1. Discussion on multi-domain pre-training and the relation to GFMs
>
> We thank the reviewer for this question. Following MDGFM, the multi-domain pre-training in Table 3 is implemented by jointly pre-training on all datasets except the downstream target dataset, selected from Cora, CiteSeer, PubMed, Cornell, Squirrel, and Chameleon.
>
> Regarding the relationship to GFMs, we clarify that GFMs aim to jointly address multi-domain pre-training and downstream transfer, focusing on how to align knowledge across source domains and how to effectively transfer the aligned knowledge to downstream tasks. Existing methods typically achieve this through domain-specific prompts, topology alignment, or mechanisms such as MoE to aggregate source-domain knowledge.
>
> In contrast, GP2F focuses on the adaptation stage. Starting from a pre-trained encoder (including GFM encoders), we explicitly model how transferable pretrained knowledge and target-specific adaptation should be balanced via a dual-branch fusion design.
>
> Importantly, our method is complementary to GFMs. The proposed fusion mechanism can be naturally applied on top of GFM encoders as a downstream adaptation module, providing a more explicit way to control the interaction between aligned source knowledge and task-specific adaptation. We will clarify this relationship in the revised version.

---

> > ### Author Rebuttal · Reviewer_tNyP · 2026-04-02
> >
> > Thanks for the author's response. All my concerns have been addressed. I will hold my positive assessment for the paper.

---

### Official Review · Reviewer_QfLZ · 2026-03-09

**Soundness:** 3
**Presentation:** 2
**Significance:** 2
**Originality:** 2
**Overall Recommendation:** 3
**Confidence:** 3

**Summary:**

This paper studies graph prompting in cross-domain few-shot settings and investigates the sources of performance gains under significant domain shift. The authors empirically observe that simple baselines such as Linear Probing (LP) and Full Fine-Tuning (FT) can achieve performance comparable to or better than existing graph prompting methods, suggesting that pretrained models already provide highly discriminative representations. To explain this phenomenon, the paper presents a theoretical analysis based on a second-order error model, showing that when estimation errors from two branches are not perfectly correlated, an affine fusion of the two estimators can achieve lower expected error than either alone. Motivated by this analysis, the authors propose a dual-branch framework that combines pretrained representations and target-domain adaptation through learnable fusion, along with contrastive alignment and a topology-consistency loss. Experiments on multiple cross-domain few-shot benchmarks demonstrate competitive performance against existing methods.

**Compliance With Llm Reviewing Policy:**

Affirmed.

**Final Justification:**

Although the authors emphasized in the rebuttal that the specific issue addressed in the paper is important, the proposed solutions are commonly used and fail to highlight the novelty of their method. I maintain my original score.

**Key Questions For Authors:**

1. On the novelty of the core observation:  The central observation of the paper is that Linear Probing (LP) and Full Fine-Tuning (FT) can often match or even outperform existing graph prompting methods on graph prompting tasks. Could the authors further clarify the main novelty of this work? In particular, does the proposed theoretical analysis lead to new design principles for graph prompt methods or provide a new task modeling perspective, rather than mainly explaining existing empirical observations?

2. On the connection between the theoretical analysis and graph prompting: The paper analyzes, from a theoretical perspective, the role of different representation sources (e.g., pretrained representations and task-specific representations) and discusses why a simple weighted combination strategy can be effective. Could the authors further clarify how this theoretical framework is specifically related to graph prompting? For example, can the theory explain why certain prompt-based methods fail to outperform LP/FT? Moreover, is there a clear correspondence between the representation fusion mechanism in the theory and existing prompt designs (e.g., feature prompts or structure prompts)?

3. On the motivation behind the method design: The proposed method is constructed by progressively introducing several components, but the overall design motivation is not sufficiently clear. Could the authors further elaborate on the rationale behind the key design choices, such as why the current modeling approach is adopted and what specific limitations of existing methods it aims to address? In addition, it would be helpful if the authors could more clearly explain the structural differences and advantages of the proposed method compared to prior work.

**Limitations:**

The discussion of limitations in the current paper is limited. In particular, the theoretical analysis relies on several assumptions (e.g., bounded variance and partially uncorrelated errors). It would be helpful to clarify under what conditions these assumptions hold and when the theory may break down under stronger distribution shifts. In addition, although the method is claimed to be parameter-efficient, the paper lacks quantitative analysis of parameter size, training cost, or inference overhead.

**Strengths And Weaknesses:**

**Strengths**

1.Well-structured and logically organized. The paper is generally well organized and easy to follow. The authors connect empirical observations with the theoretical model in Section 3, which provides motivation for the method design in Section 4. This results in a relatively coherent transition from empirical findings to theoretical analysis and finally to model instantiation.

2.Comprehensive experimental evaluation. The proposed method is evaluated on 15 benchmarks, including 9 node classification datasets and 6 graph classification datasets. The experimental results are largely consistent with the theoretical analysis and demonstrate the robustness of the proposed approach under 1-shot, 3-shot, and 5-shot settings.

**Weaknesses**

1.Limited novelty of the core insight. The paper attributes the performance gains of graph prompting to three factors: general knowledge encoded in pretrained models, task-specific knowledge introduced during target-domain adaptation, and their fusion. However, this perspective has been widely discussed in the literature on transfer learning and domain adaptation. Providing a more task-specific perspective tailored to graph prompting could further strengthen the novelty of the work.

2.Theoretical analysis could be more tailored to graph prompting. The paper shows that combining two estimators using a second-order error model can outperform each individual branch and potentially reduce the classification error bound. While the result is reasonable, the analysis is relatively generic and does not explicitly incorporate characteristics of graph structures. A deeper discussion connecting the theoretical findings to graph-specific properties could make the analysis more insightful.

3.Motivation behind the method design could be clarified. (1) The GP2F framework integrates lightweight adapters after each convolution layer, but the paper does not provide clear theoretical or empirical justification for why layer-wise adaptation would be preferable to applying adaptation only at the encoder output.
(2) The dual-branch fusion paradigm has been explored in prior graph prompting works. A clearer discussion of how this design differs from existing approaches and what advantages it brings would help highlight the contributions.

4.Experimental analysis could be further strengthened.Although the paper argues that dual-branch fusion reduces representation estimation error and tightens the classification error bound, the experimental section mainly focuses on performance metrics. Additional qualitative analyses, such as representation visualization, could provide more intuitive insights. Moreover, while the paper mentions parameter-efficient tuning, quantitative analysis of parameter efficiency or computational cost is currently limited.

5.The framework figure could be improved. In the current figure, the two branches appear visually very similar, and depicting the second branch as a frozen encoder may lead to confusion. In addition, some inconsistencies in font size and layout reduce readability.

6.Related work could be strengthened. While the paper reviews several related methods, the discussion mainly summarizes existing work without clearly analyzing their respective strengths and limitations. A more structured comparison could help readers better understand the differences between the proposed method and prior approaches.

---

> ### Author Rebuttal · Authors · 2026-03-30
>
> Thanks for your valuable remarks. W4/W5 repository is available at https://anonymous.4open.science/r/GP2F-0357/. Responses are below. All the experiments and discussions will be included in the future version.
>
> # W1/Q1
>
> Thanks for the helpful comment and question. We would like to clarify that our goal is specific to cross-domain graph prompting, rather than a generic transfer-learning perspective. Existing graph prompting methods usually treat prompting as the main adaptation mechanism on top of a pretrained graph encoder. In contrast, our empirical study explains an underexplored phenomenon: under domain shift, LP and FT are already highly competitive baselines. This suggests that the key challenge is not merely to design a well prompt module, but to explicitly balance transferable pretrained graph knowledge and target-specific adaptation. Therefore, our value is not to restate a generic fusion perspective, but to identify this underexplored graph-prompting-specific issue and derive a more suitable design principle, instantiated by GP2F.
>
> # W2/Q2
>
> Thanks for the helpful comment and question. We agree that the theory in Sec. 3 does not explicitly model graph structures, and is therefore not specific to graph prompting. Its role in our paper is instead to explain why, in cross-domain graph prompting, prompting-based adaptation does not always consistently outperform LP/FT: pretrained and task-adapted representations often remain useful in different ways under domain shift. This motivates a simple principle: explicitly preserving and combining both sources can be more effective than relying on only one. Under this view, feature/structure/adapter-style prompts can all be regarded as different ways of constructing the adapted representation, while LP is closer to relying on the pretrained representation. We will revise the paper to clarify this scope and make it clearer that the theorem itself is not specific to graph prompting.
>
> # W3-1
>
> Thanks. In cross‑domain scenario, the discrepancy can accumulate across layers. Hence, layer‑wise adapters allow progressive adjustment across layers. We validate this design with an ablation as follows:
>
> |             | Cora       | CiteSeer    | PubMed      | Computers   | Photo       | CS         | WikiCS     |
> | ----------- | ---------- | ----------- | ----------- | ----------- | ----------- | ---------- | ---------- |
> | output-only | 56.38±9.13 | 50.95±11.12 | 52.96±11.12 | 54.46±10.37 | 67.23±11.10 | 66.70±7.60 | 46.29±9.01 |
>
> The results show better performance of layer-wise adaptation.
>
> # W3-2/Q3
>
> Thanks for the question. Although dual-branch paradigms have appeared in prior work (e.g., GraphControl[1] and GraphLoRA[2]), their motivations and mechanisms differ fundamentally from ours.
>
> GraphControl adapts downstream features via a GCC-based fine-tuning. Turning features to the GCC input format, it generate condition input and add it to frozen representations to provide feature knowledge. GraphLoRA tunes a fully learnable LoRA branch whose output is added to the frozen representations for structural adaptation, and relies on SMMD to align feature distributions, thus requiring source-domain data.
>
> In contrast, GP2F explicitly addresses how to fuse general and task-specific knowledge in representation level, yielding a universal framework applicable to different pre-trained models in source-free settings.
> # W4
> Thanks. We conducted 5‑shot t‑SNE visualizations of GP2F and three representative baselines. GP2F shows clearer inter‑class boundaries.
> We report results on parameter as follows:
> | Method          | GRACE(LP) | GRACE(FT) | GPF  | EdgePrompt | GPPT  | GraphPrompt | DAGPrompT | GraphLoRA | Ours |
> | --------------- | :-------: | :-------: | :--: | :--------: | :---: | :---------: | :-------: | :-------: | :--: |
> | trainable/FT(%) |   0.96    |  100.00   | 1.67 |    1.73    | 15.21 |    0.06     |   8.06    |   29.87   | 9.23 |
>
> Train/inference time and GPU memory in link and **W2 of Reviewer z2mL**. The results show that GP2F incur modest overhead while providing notable performance gains.
> # W5
> Thanks. The figure has been updated: encoder renamed and recolored, fonts standardized.
> # W6
> Thanks for the suggestion. A brief summary below and full in final version.
> In-domain GPL methods handle pre-training vs downstream misalignment well but focus only on task adaptation and often fail cross-domain.
> Graph foundation models capture source knowledge upstream and generate domain prompts downstream (e.g., via MoE or weighted aggregation). Downstream is typically single-pathway, so task-specific and domain prompts interact only implicitly. Cross-domain GPL methods have been discussed in Q3.
> # Limitation
> Please refer to **W1 of Reviewer Z3PJ** and W4.
>
> [1]Graphcontrol: Adding conditional control to universal graph pre-trained models for graph domain transfer learning. WWW24.
>
> [2]Graphlora: Structure-aware contrastive low-rank adaptation for cross-graph transfer learning. KDD25.

---

> > ### Author Rebuttal · Reviewer_QfLZ · 2026-04-02
> >
> > Cross-domain learning is a branch of Transfer Learning. Therefore, positioning cross-domain graph prompting as conceptually distinct from transfer learning does not fully resolve the concern regarding novelty. I will maintain my score.

---

> > > ### Author Response · Authors · 2026-04-02
> > >
> > > Thanks for your comment. We understand that cross-domain graph prompting belongs to the broader family of transfer learning. However, our point is not to position GPL as conceptually separate from transfer learning, but to clarify the scope and motivation of our contribution. Our method is motivated by a phenomenon that is clearly observed in cross-domain graph prompting: under domain shift, LP and FT remain highly competitive, which means that prompting should not be treated only as a single adaptation pathway. This is the specific issue we aim to address.
> > >
> > > More importantly, this motivation is established in GPL, but does not necessarily extend equally to broader transfer learning settings with different assumptions on source access, supervision, and adaptation protocols. Therefore, our novelty is not a general claim about transfer learning, but a graph-prompting-specific problem formulation and design principle grounded in the GPL setting.

---

### Official Review · Reviewer_z2mL · 2026-03-10

**Soundness:** 3
**Presentation:** 3
**Significance:** 3
**Originality:** 3
**Overall Recommendation:** 5
**Confidence:** 4

**Summary:**

In this paper, the authors study cross-domain graph prompt learning. Specifically, the authors first show that two simple baselines, full fine-tuning and linear probing, show competitive results, and theoretically demonstrate that jointly optimizing a frozen branch and an adapted branch outperforms optimizing either branch alone. Based on these findings, the paper proposes GP2F, a dual-branch method for cross-domain graph prompt learning, incorporating a frozen branch to retain pre-trained knowledge and an adapted branch for task-specific adaptation. Extensive experiments show that the proposed method achieves state-of-the-art performance across multiple baselines and benchmarks.

**Compliance With Llm Reviewing Policy:**

Affirmed.

**Final Justification:**

The authors‘ responses have addressed my comments. I maintain my positive score. I believe this paper makes a solid contribution towards graph prompt learning.

**Key Questions For Authors:**

Please refer to the weaknesses identified above. No additional questions.

**Limitations:**

Yes

**Strengths And Weaknesses:**

Strengths:
S1. The paper provides an interesting empirical finding for graph prompt learning that simple baselines such as linear probing and full fine-tuning remain highly competitive in cross-domain settings, which provides useful insights for future research on graph prompt learning.
S2. The paper establishes a clear connection between theory and methodological design. The theoretical analysis explains why combining a frozen branch and an adapted branch can reduce estimation error, which directly motivates the proposed dual-branch architecture and fusion strategy.
S3. The experiments are comprehensive, spanning multiple pre-training strategies and several datasets across node and graph classification.

Weaknesses:
W1. The method section needs clarification. For example, the correspondence between \lambda in the theoretical analysis and \alpha in the method section should be made explicit. The relationship between the BCE loss shown in Fig. 2 and L_{fus} is not clearly stated. The use of the projector in the baselines should also be clarified.
W2. The paper currently lacks an efficiency analysis. Since GP2F introduces an additional branch and fusion mechanism, it may incur extra runtime or memory overhead compared with simpler baselines. It would be helpful to report such costs for a clearer understanding of its practical efficiency.
W3. The paper does not clearly distinguish between cross-domain transfer and cross-dataset transfer in its preliminary. While the experiments involve different datasets, a clearer explanation of how domain shift is defined and controlled in each setting would improve conceptual clarity.
W4. Experiments with different pre-training datasets would further strengthen the empirical evaluation.

---

> ### Author Rebuttal · Authors · 2026-03-30
>
> We are thankful for the reviewer’s support of our work and insightful suggestions that have helped us strengthen our work. We respond to each comment as follows.
>
> # W1
>
> Thanks. $\lambda$ in the theoretical analysis corresponds to $\alpha$ in the implementation and the BCE loss in Fig. 2 corresponds to $\mathcal{L}_\mathrm{fus}$, which is instantiated as BCE. We will clarify this in the text and figure. The projector is a single-layer MLP, applied consistently across all methods for fair comparison. We will clarify this in the final version.
>
> # W2
>
> Thanks. We conducted quantitative analysis on training time and GPU memory as follows:
>
> | Method\|Time(ms/epoch) | Cora | CiteSeer | PubMed | Computers | Photo | CS    | WikiCS |
> | ---------------------- | ---- | -------- | ------ | --------- | ----- | ----- | ------ |
> | GRACE(LP)              | 2.9  | 3.5      | 3.6    | 6.0       | 3.6   | 14.7  | 5.0    |
> | GRACE(FT)              | 3.2  | 3.9      | 4.4    | 7.1       | 4.9   | 15.2  | 5.1    |
> | GPF                    | 3.0  | 3.5      | 3.6    | 6.1       | 4.0   | 14.5  | 5.2    |
> | EdgePrompt             | 2.5  | 2.8      | 6.8    | 23.1      | 12.3  | 20.5  | 20.6   |
> | GPPT                   | 6.0  | 5.7      | 4.9    | 10.4      | 7.3   | 21.9  | 9.7    |
> | GraphPrompt            | 5.0  | 4.9      | 4.9    | 6.4       | 5.7   | 8.9   | 5.6    |
> | DAGPrompT              | 10.7 | 8.5      | 8.6    | 12.5      | 17.7  | 16.3  | 12.0   |
> | GraphLoRA              | 20.1 | 34.3     | 451.7  | 232.3     | 86.2  | 403.7 | 172.5  |
> | Ours                   | 7.6  | 8.8      | 165.5  | 88.6      | 31.4  | 154.4 | 67.5   |
>
> | Method\|Space (MB) | Cora | CiteSeer | PubMed | Computers | Photo | CS    | WikiCS |
> | ------------------ | ---- | -------- | ------ | --------- | ----- | ----- | ------ |
> | GRACE(LP)          | 94   | 186      | 334    | 711       | 370   | 942   | 594    |
> | GRACE(FT)          | 96   | 188      | 335    | 712       | 372   | 944   | 596    |
> | GPF                | 109  | 204      | 442    | 787       | 412   | 1043  | 659    |
> | EdgePrompt         | 223  | 288      | 1423   | 5949      | 2914  | 2804  | 5190   |
> | GPPT               | 94   | 185      | 334    | 711       | 370   | 941   | 594    |
> | GraphPrompt        | 644  | 925      | 2338   | 11003     | 5754  | 13241 | 4161   |
> | DAGPrompT          | 399  | 925      | 1014   | 1284      | 715   | 13749 | 461    |
> | GraphLoRA          | 469  | 718      | 18177  | 10363     | 3615  | 16604 | 7815   |
> | Ours               | 533  | 830      | 22739  | 11241     | 3596  | 20237 | 8161   |
>
> The results show that GP2F incur modest overhead while providing notable performance gains. We will include these results in the final version.
>
> # W3
>
> Thanks for this helpful question. Cross-dataset transfer is treated as a practical form of cross-domain transfer, as methods like GRACE are not designed for multi-domain settings. Since our benchmarks cover citation, co-purchase, co-authorship, and webpage networks, they provide an effective testbed for cross-domain generalization.
>
> # W4
>
> Thanks. We conducted experiments pretrained on PubMed and Computers. The results are as follows:
>
> | Method(PubMed) | Cora          | CiteSeer      | PubMed        | Computers     | Photo         | CS            | WikiCS       |
> | -------------- | ------------- | ------------- | ------------- | ------------- | ------------- | ------------- | ------------ |
> | GRACE(LP)      | 34.85 ± 9.87  | 28.13 ± 6.72  | 50.84 ± 8.96  | 52.13 ± 12.92 | 62.50 ± 12.16 | 58.58 ± 10.50 | 42.06 ± 8.51 |
> | DAGPrompT      | 48.47 ± 8.85  | 40.58 ± 8.61  | 50.90 ± 9.61  | 37.64 ± 10.95 | 51.96 ± 8.36  | 65.88 ± 10.01 | 30.05 ± 7.10 |
> | GraphLoRA      | 50.11 ± 10.81 | 37.95 ± 10.92 | 50.32 ± 9.36  | 51.64 ± 11.78 | 64.59 ± 11.46 | 43.74 ± 12.76 | 38.91 ± 8.55 |
> | Ours           | 55.41 ± 8.73  | 48.93 ± 11.42 | 53.16 ± 10.00 | 52.79 ± 10.28 | 66.96 ± 10.33 | 64.47 ± 8.58  | 45.93 ± 8.71 |
>
> | Method(Computers) | Cora          | CiteSeer      | PubMed        | Computers     | Photo         | CS            | WikiCS       |
> | ----------------- | ------------- | ------------- | ------------- | ------------- | ------------- | ------------- | ------------ |
> | GRACE(LP)         | 34.65 ± 9.58  | 29.01 ± 6.42  | 51.05 ± 8.61  | 53.22 ± 12.26 | 60.75 ± 10.89 | 59.30 ± 10.41 | 42.09 ± 9.12 |
> | DAGPrompT         | 50.46 ± 9.37  | 46.53 ± 9.27  | 51.29 ± 9.65  | 40.62 ± 11.83 | 54.17 ± 8.75  | 68.06 ± 8.81  | 30.45 ± 6.90 |
> | GraphLoRA         | 47.58 ± 10.81 | 33.71 ± 10.40 | 47.94 ± 12.75 | 52.78 ± 11.74 | 65.63 ± 11.27 | 41.47 ± 12.05 | 40.12 ± 8.37 |
> | Ours              | 59.08 ± 9.47  | 50.78 ± 12.05 | 53.64 ± 11.56 | 54.43 ± 10.70 | 66.86 ± 10.68 | 68.48 ± 8.17  | 44.82 ± 7.61 |
>
> The results show that GP2F consistently outperforms baselines. These results will be included in the final version. Full results in the repository https://anonymous.4open.science/r/GP2F-0357/ due to character limitation.

---

> > ### Author Rebuttal · Reviewer_z2mL · 2026-04-02
> >
> > The authors‘responses’ have addressed my comments. I maintain my positive score.

---

### Official Review · Reviewer_Z3PJ · 2026-03-13

**Soundness:** 3
**Presentation:** 3
**Significance:** 3
**Originality:** 2
**Overall Recommendation:** 4
**Confidence:** 3

**Summary:**

The paper studies cross-domain graph prompt learning for adapting pre-trained GNNs under distribution shift. The authors theoretically show that combining representations from a frozen encoder and an adapted encoder can reduce estimation error. Based on this insight, they propose GP2F, a dual-branch framework that fuses a frozen pre-trained branch with an adapter-based adapted branch via a learnable weight, along with contrastive alignment and topology-consistent fusion losses. Experiments on multiple cross-domain node and graph classification benchmarks demonstrate improved performance over existing graph prompt learning methods.

**Compliance With Llm Reviewing Policy:**

Affirmed.

**Final Justification:**

The authors have addressed my concerns. I will keep my score.

**Key Questions For Authors:**

1. The theoretical analysis assumes that the errors of the two encoders are unbiased. In practice, however, the frozen encoder and the adapted encoder are trained under different objectives and may exhibit systematic bias under domain shift. Could the authors discuss whether this assumption is realistic and how sensitive the method is when this assumption does not hold?

2. Figure 5 shows that the performance depends on the adapter projection dimension $r$. Could the authors provide guidance on how to select $r$ in practice, and whether the method is sensitive to this hyperparameter across different datasets?

3. Could the authors provide further analysis on how much the adapted branch actually contributes compared to the frozen branch in extremely low-shot settings (e.g., 1-shot)?

**Limitations:**

yes

**Strengths And Weaknesses:**

Strength:

1. The paper studies cross-domain graph prompt learning, where the pre-training data and downstream tasks come from different graph distributions. This setting is practically relevant in many real-world applications.

2. The authors show that in cross-domain settings, simple baselines such as linear probing and full fine-tuning can achieve performance comparable to, or even competitive with, existing graph prompt learning methods. This observation motivates a deeper investigation into the mechanism of prompt-based adaptation.

3. The proposed GP2F framework adopts a dual-branch architecture that combines a frozen pre-trained encoder with an adaptable encoder, allowing the model to preserve pre-trained knowledge while performing task-specific adaptation. The overall design is conceptually simple and easy to implement.

4.  Experiments are conducted on multiple node classification and graph classification datasets, including both few-shot and large-scale settings. The results demonstrate consistent performance improvements over several baseline methods.

Weakness:

1. The theoretical analysis relies on assumptions such as latent linear separability and unbiased noise in the representations produced by the frozen and adapted encoders. These assumptions may be difficult to satisfy in practical graph representation learning scenarios.

2. Although the paper focuses on cross-domain scenarios, the method does not explicitly quantify or model the distribution discrepancy between the source and target graphs. It is unclear how the proposed approach adapts to different types or magnitudes of domain shifts.

---

> ### Author Rebuttal · Authors · 2026-03-30
>
> Thanks for your support of our work and the insightful and valuable feedback. Result of Q1 is in https://anonymous.4open.science/r/GP2F-0357/To_Reviewer_Z3PJ/Q1_result.pdf. Our detailed responses are provided below.
>
> # W1. Response to concerns on theoretical assumptions in practice
>
> We thank the reviewer for this question. We agree that assumptions such as latent linear separability and unbiased estimation errors are introduced to provide an idealized setting for analytical simplicity. However, our goal is not to precisely model a specific practical method, but rather to offer a principled perspective on knowledge fusion in cross-domain graph prompt learning.
>
> Specifically, under frozen and adapted representations, the analysis shows that when the two branches act as noisy estimators of the ideal representation $z_{i}$, their fusion can reduce estimation error compared with either branch alone. Therefore, the theoretical contribution is intended to explain why combining pretrained knowledge with target-specific adaptation can be beneficial. Although our assumptions do not fully match practical settings, the resulting conclusion is still helpful and motivates the core design of GP2F.
>
> The analysis relies on the observation that both branches can reasonably estimate $z_i$, which is supported by the strong performance of LP and FT as baselines. When either branch yields a poor estimate, the theoretical conditions (e.g., unbiasedness or error independence) may break down. For example, the frozen branch lacks sufficient general knowledge or the adapted branch overfits. In such cases, the benefit of fusion may diminish.
>
> # Q1. Response to “unbiased errors” in Assumption 3.2
>
> Thanks for this question. We would like to clarify that the “unbiased errors” in Assumption 3.2 is a modeling assumption used for analytical simplicity, and it is not meant to claim the absence of domain shift. Both $h^g_i$ and $h^a_i$ are node representations of the same target-domain and are viewed as two noisy estimates of the same latent target representation $z_{i}$. The zero-mean assumption is introduced for analytical simplicity, allowing us to isolate the roles of variance and cross-covariance in fusion. Since both branches are built on the same backbone and operate on the same target-domain data, we view it as a reasonable approximation for theory rather than a claim that domain shift or systematic discrepancy does not exist. We will revise the paper to better clarify this interpretation.
>
> For the sensitivity of GP2F to domain shift, please refer to the experiment across different pretrain datasets (**W2 of Reviewer z2mL**) or the link above.
>
> # W2 and Q3. Response to concerns on domain shift adaptation and branch contribution
>
> We thank the reviewer for this insightful comment. We clarify that, in cross-domain graph prompt learning, source-domain graphs are unavailable during downstream adaptation, making it infeasible to explicitly quantify the source–target discrepancy. Nevertheless, source-domain knowledge is implicitly encoded in the pretrained encoder. GP2F adapts to different levels of domain shift through the adaptive fusion of pretrained knowledge and task-adaptive knowledge. Specifically, this balance is controlled by the fusion weight $\alpha$, which adapts during training and indicate the relative dependence on each branch under different levels of domain shift. When pre-trained on Cora and transferred to similar citation networks (e.g., CiteSeer and PubMed), $\alpha$ shows a clear downward trend during training, indicating that the adapted branch becomes increasingly reliable and contributes more as optimization proceeds. For datasets with larger domain shifts (e.g., Computers, Photo, CS, and WikiCS), we observe that $\alpha$ first rises slightly in the early stage of training, suggesting that the model initially relies more on the frozen branch when the randomly initialized adapted branch is not yet reliable. As training progresses, the contribution of the adapted branch gradually increases, and $\alpha$ gradually decreases. Overall, $\alpha$ decrease to about 0.4 to 0.2 across datasets.
>
> These observations show that $\alpha$ behaves differently under varying levels of domain shift, revealing how GP2F adapts to cross-domain discrepancies. We will clarify this analysis in the final version.
>
> # Q2. Discussion on the choice of hyperparameter $r$
>
> Thanks for this question. We provide the analysis of $r$ in Line 405, Fig. 5, Fig. 7, and Fig. 8. Overall, the method is not highly sensitive to $r$ across different datasets. In practice, $r$ is determined by both the dataset scale and the magnitude of cross-domain discrepancy, since it controls the capacity of the adaptation branch to encode task-specific corrections. We generally recommend starting with 8 or 16, and then adjusting it if needed.

---

> > ### Author Rebuttal · Reviewer_Z3PJ · 2026-04-02
> >
> > The authors have addressed my concerns. I will keep my score.

---

### Decision · Program_Chairs · 2026-04-30

**Decision:**

Accept (regular)

**Comment:**

This paper proposes GP2F, a dual-branch framework that fuses frozen and adapted representations for cross-domain graph prompt learning. The work is well-motivated and addresses a practically important setting. A key strength is the clear connection between empirical observations, theory, and model design. The proposed method is effective, and extensive experiments across diverse benchmarks consistently demonstrate strong performance improvements over competitive baselines. While some clarifications on theory, efficiency, and presentation would strengthen the paper, reviewers are generally positive about this work and believe these issues can be addressed in a revised version.